# Data-driven optimisation of wind farm layout and wake steering with large-eddy simulations

Nikolaos Bempedelis[1], Filippo Gori[1], Andrew Wynn[1], Sylvain Laizet[1], and Luca Magri[1,2,3]

[1]Department of Aeronautics, Imperial College London, SW7 2AZ London, UK
[2]The Alan Turing Institute, London NW1 2DB, UK
[3]Politecnico di Torino, DIMEAS, Corso Duca degli Abruzzi, 24 10129 Torino, Italy

**Correspondence:** Nikolaos Bempedelis (n.bempedelis20@imperial.ac.uk); Luca Magri (l.magri@imperial.ac.uk)

**Abstract.** Maximising the power production of large wind farms is key to the transition towards net zero. The overarching goal of this paper is to propose a computational method to maximise the power production of wind farms with two practical design strategies. First, we propose a gradient-free method to optimise the wind farm power production with high-fidelity surrogate models based on large-eddy simulations and a Bayesian framework. Second, we apply the proposed method to maximise wind farm power production by both micro-siting (layout optimisation) and wake steering (yaw angle optimisation). Third, we compare the optimisation results with the optimisation achieved with low-fidelity wake models. Finally, we propose a simple multi-fidelity strategy by combining the inexpensive wake models with the high-fidelity framework. The proposed gradient-free method can effectively maximise wind farm power production. Performance improvements relative to wake-model optimisation strategies can be attained, particularly in scenarios of increased flow complexity, such as in the wake steering problem, in which some of the assumptions in the simplified flow models become less accurate. The optimisation with high-fidelity methods takes into account nonlinear and unsteady fluid mechanical phenomena, which are leveraged by the proposed framework to increase the farm output. This paper opens up opportunities for wind farm optimisation with high-fidelity methods and without adjoint solvers.

## 1 Introduction

The transition towards a net-zero world is intertwined with a continually growing demand for renewable energy sources. Wind power has emerged as the leading contributor to renewable electricity in numerous countries (Our World in Data, 2022). Governments worldwide have recognised the significance of this resource and have committed to substantial capacity expansions as part of their energy strategies. As wind energy takes on an increasingly central role as an energy source, it is critical for it to mature in terms of efficiency and resilience. This entails addressing a number of challenges, which range from turbine aerodynamics, their interaction with the atmosphere, and plant-level control (Veers et al., 2019, 2022).

One key issue is that wind farms that consist of many turbine rows are typically less efficient than less deep wind farms (i.e., downstream rows produce less power than upstream ones) (Barthelmie et al., 2009). This degradation in efficiency is primarily attributed to wake losses, which arise when a significant portion of the turbines operate within the wake fields generated by neighbouring turbines within the same farm. A wind turbine operating within a wake field is a negative operability issue for

two reasons. First, the reduction of its power output due to the decelerated incoming wind, and, second, the increase of fatigue loading due to higher levels of turbulence.

The long-established strategy for mitigating wake losses is to optimally position the wind turbines across the available land. This approach is most commonly referred to as *layout optimisation* or *micro-siting*. Building upon the work by Mosetti et al. (1994), numerous studies have been published in the literature on maximising the production (or minimising the cost of energy production) of a wind farm by optimising turbine placement (see, for instance, Grady et al. (2005); Kusiak and Song (2010); Chowdhury et al. (2012); Stanley and Ning (2019), and the review articles by Herbert-Acero et al. (2014) and Shakoor et al. (2016)).

More recently, an approach called *wake steering* was proposed as an alternative strategy to mitigate wake losses. In wake steering, upstream turbines are yawed to deflect their wakes away from downstream turbines. Although this results in a reduction in power production from the upstream turbines, the downstream turbines generate more power, leading to an overall increase in the farm's power output. Demonstrations of performance gains in a number of computational (Fleming et al., 2015; Gebraad et al., 2016), experimental (Adaramola and Krogstad, 2011; Campagnolo et al., 2016; Bastankhah and Porté-Agel, 2019) and field (Fleming et al., 2017; Howland et al., 2019; Fleming et al., 2019; Simley et al., 2021) studies have prompted the consideration of implementing wake steering in commercial wind plants (see, for instance, a press release by WindE-SCo (2023)). Nevertheless, further research is required in order to reduce the uncertainties regarding its potential benefits (Kheirabadi and Nagamune, 2019).

In the vast majority of relevant studies published in the literature, the optimisation of the farm layout or the turbine yaw angles is carried out with low-fidelity flow solvers, commonly known as *wake models*, as cost function evaluators. Wake models are tools that estimate the velocity deficit downstream of a wind turbine and, by means of superposition, the overall flow in a wind farm (see, for example, Jensen (1983); Frandsen et al. (2006); Bastankhah and Porté-Agel (2014); Bempedelis and Steiros (2022), and the review article by Porté-Agel et al. (2020)). Wake models are built on simplified assumptions for the turbulent wake of a porous disk and do not account for various phenomena such as unsteadiness, non-linear interactions, or blockage, among others. As a consequence, optimisation based on wake models misses opportunities to improve farm performance by manipulating and potentially exploiting these unaccounted phenomena. Furthermore, wake models can produce predictions which are non-smooth or discontinuous as functions of the design variables (e.g., turbine coordinates or yaw angles). This renders their use within gradient-based optimisation algorithms challenging (Gori et al., 2023). Nevertheless, wake models are almost invariably used in optimisation studies due to their low computational cost, and the expertise gained from decades of development and application (Asmuth et al., 2023).

On the other hand, higher fidelity flow models have been employed in a limited number of layout optimisation studies (King et al., 2017; Antonini et al., 2018; Allen et al., 2020; Antonini et al., 2020). In these works, wind farm layouts were optimised with an adjoint gradient-based approach and steady-state Reynolds-averaged Navier-Stokes (RANS) simulations. The use of steady RANS enables the consideration of a number of the aforementioned phenomena, such as blockage or pressure effects. However, the unsteady wake dynamics and the atmosphere-to-wake interactions, which play a critical role in large wind farm flows, cannot be appropriately accounted for. While higher-fidelity flow models, such as large-eddy simulations (LES), aptly

take these phenomena into account, the computation of the gradient of a long-term[1] time average of a chaotic turbulent flow is ill posed (Wang, 2013; Blonigan and Wang, 2018; Huhn and Magri, 2022), which is a limit for adjoint methods. Despite these challenges, large-eddy simulations and adjoint methods have been combined in the context of wind farm control (Goit and Meyers, 2015; Munters and Meyers, 2018), showcasing the potential of dynamic control strategies. Alternatively, LES have been utilised in hybrid or multi-fidelity approaches (Bokharaie et al., 2016; Kirby et al., 2023). For instance, in Bokharaie et al. (2016), LES was employed to calibrate the parameters of wake models, which, in turn, were used to find optimal wind farm layouts.

In this work, we propose a data-driven framework for optimising the output of a wind farm based on a gradient-free Bayesian approach and high-fidelity large-eddy simulations of the wind farm flow. The structure of the paper is as follows. The proposed optimisation strategy is outlined in Sect. 2. Section 3 discusses its application to a sixteen-turbine layout optimisation problem. In Sect. 4, the framework is applied to a ten-turbine yaw angle optimisation (wake steering) problem. Finally, conclusions end the paper in Sect. 5.

## 2 Methodology

The methodology is presented in two parts. First, we introduce the optimisation framework in Sect. 2.1. Second, in Sect. 2.2, we describe the flow solver employed to predict the wind farm flow and power output.

### 2.1 Bayesian optimisation

Bayesian optimisation (BO) is a gradient-free global optimisation strategy, which is particularly attractive for optimising complex functions (as in large codes for which the adjoint algorithm is cumbersome). Its effectiveness has been demonstrated in various fluid mechanical applications (e.g., Mahfoze et al. (2019); Huhn and Magri (2022); O'Connor et al. (2023)). It seeks for the extrema of the objective function by constructing a probabilistic surrogate model of it, typically a Gaussian process (GP), and by exploring the parameter space with an acquisition function (AF).

A Gaussian Process (GP) model is a non-parametric probabilistic model that is defined by a prior mean function, $\mu_0(\boldsymbol{x})$, $\boldsymbol{x} \in \mathbb{R}^d$, which is usually assumed zero, and a covariance function (also known as the kernel), $k \in \mathbb{R}(\boldsymbol{x}, \boldsymbol{x}')$. Given a set $\mathcal{D}$ of $m$ observations $\boldsymbol{Y} \in \mathbb{R}^{m \times 1} = f(\boldsymbol{X}) = [y_1, \ldots, y_m]^\mathsf{T}$ at the input locations $\boldsymbol{X} \in \mathbb{R}^{m \times d} = [\boldsymbol{x}_1, \ldots, \boldsymbol{x}_m]^\mathsf{T}$, as training data, and assuming a zero prior mean, we can compute the posterior mean, $\boldsymbol{\mu}_* \in \mathbb{R}^{n \times 1}$, and variance, $\boldsymbol{\sigma}_*^2 \in \mathbb{R}^{n \times 1}$, for a set of $n$ test points $\boldsymbol{X}_* \in \mathbb{R}^{n \times d}$ as

$$\boldsymbol{\mu}_* = \boldsymbol{K}_*^T \left( \boldsymbol{K} + \sigma^2 \boldsymbol{I} \right)^{-1} \boldsymbol{Y} \tag{1}$$

$$\boldsymbol{\sigma}_*^2 = \mathrm{diag} \left( \boldsymbol{K}_{**} - \boldsymbol{K}_*^T \left( \boldsymbol{K} + \sigma^2 \boldsymbol{I} \right)^{-1} \boldsymbol{K}_* \right) \tag{2}$$

where $\boldsymbol{K} \in \mathbb{R}^{m \times m} = k(\boldsymbol{X}, \boldsymbol{X})$, $\boldsymbol{K}_* \in \mathbb{R}^{m \times n} = k(\boldsymbol{X}, \boldsymbol{X}_*)$, $\boldsymbol{K}_{**} \in \mathbb{R}^{n \times n} = k(\boldsymbol{X}_*, \boldsymbol{X}_*)$, and $\sigma^2 \in \mathbb{R}$ is the observational noise. These expressions provide an estimate of the underlying objective function and the corresponding uncertainty for ev-

---

[1]By "long-term" we mean after many Lyapunov times.

ery point in the parameter space. *Training* a GP model involves determining the values of the kernel hyper-parameters that maximise the marginal likelihood. For a more detailed discussion on GPs, the reader is referred to Rasmussen and Williams (2006).

The acquisition function $\text{AF}(\boldsymbol{x})$, which is built on the surrogate model whose first two statistical moments are given by equations (1)-(2), determines the next point(s) to be evaluated, by taking into account both the mean and uncertainty for any point $\boldsymbol{x}$ of the parameter space, and providing the probability, or amount, by which $\boldsymbol{x}$ can improve the current optimum. Exploration, which directs the search towards unexplored areas of the parameter space, and exploitation, which focuses the search in the vicinity of known promising solutions, are balanced based on the trade-off between uncertainty reduction and potential improvement. In brief, Bayesian optimisation is described in Algorithm 1.

---

**Algorithm 1** Bayesian Optimisation

---

1: Acquire a set of initial objective function observations $\mathcal{D}$
2: Initialise the variables holding the best solution, $\boldsymbol{x}_{\text{opt}}$ and $y_{\text{opt}}$
3: **for** $i = 1$ **to** stopping criterion **do**
4:     *// Update the surrogate model using the available data*
5:     Train the GP model given $\mathcal{D}$
6:     *// Find the next point to evaluate using the AF*
7:     $\boldsymbol{x}_{\text{next}} \leftarrow \arg\max \text{AF}(\boldsymbol{x})$
8:     *// Evaluate the objective function at $\boldsymbol{x}_{next}$*
9:     $y_{\text{next}} \leftarrow f(\boldsymbol{x}_{\text{next}})$
10:    *// Update the dataset with the new evaluation*
11:    Append $\boldsymbol{x}_{\text{next}}$ and $y_{\text{next}}$ to $\mathcal{D}$
12:    *// Update the best solution found so far*
13:    **if** $y_{\text{next}} > y_{\text{opt}}$ **then**
14:       $\boldsymbol{x}_{\text{opt}} \leftarrow \boldsymbol{x}_{\text{next}}$
15:       $y_{\text{opt}} \leftarrow y_{\text{next}}$
16:    **end if**
17: **end for**
18: **return** $\boldsymbol{x}_{\text{opt}}, y_{\text{opt}}$

---

In the current work, Bayesian optimisation is implemented with the `GPyOpt` library (The GPyOpt authors, 2016). As discussed in more detail in Sect. 3.2, we consider GP priors characterised by the Rational Quadratic kernel (Rasmussen and Williams, 2006). We further assume that our LES observations are high-fidelity, and, as such, noise-free (i.e., $\sigma^2$ is negligible). To select the next observation point(s), we use the Lower Confidence Bound acquisition function, $\text{AF}(\boldsymbol{x}) = \mu_*(\boldsymbol{x}) - \lambda\sigma_*(\boldsymbol{x})$, which allows for straightforward control of the exploration-to-exploitation ratio through the tunable parameter $\lambda$, which weighs the mean/exploitation $\mu_*$ and uncertainty/exploration $\sigma_*$ terms of the GP model, with larger values of $\lambda$ favouring exploration. Other AFs (e.g., Probability of Improvement, Expected Improvement) or combination of them may be used (e.g., Huhn and

Magri (2022)). The kernel hyper-parameter and AF optimisations are carried out with the L-BFGS algorithm (Liu and Nocedal, 1989). Finally, we compute design points to be explored in batches (batch BO), following the local penalisation method proposed in González et al. (2016). This allows us to efficiently utilise our high-performance computing resources by running several LES at the same time.

## 2.2 Large-eddy simulations

To predict the wind farm flow and power output, we employ the wind farm simulator `Winc3d` (Deskos et al., 2020) of the open-source[2] finite-difference framework `Xcompact3d` (Bartholomew et al., 2020). `Xcompact3d` solves the incompressible filtered Navier-Stokes equations on Cartesian meshes using sixth-order compact schemes and a third-order Adams-Bashforth method for time integration (Laizet and Lamballais, 2009). Efficient scaling up to hundreds of thousands of CPU cores is made possible by utilising the `2Decomp & FFT` library for parallelisation (Laizet and Li, 2011). `Winc3d` has been validated against experimental and field data in several wind turbine and wind farm flow problems (Deskos et al., 2019, 2020; Bempedelis and Steiros, 2022; Steiros et al., 2022; Jané-Ippel et al., 2023).

In this work, the Smagorinsky sub-filter scale model (Smagorinsky, 1963) is used to account for the effects of the unresolved fluid motions. The wind turbines are modelled with the non-rotational actuator disk method, which is described in more detail, together with validation studies in Bempedelis et al. (2023) and Jané-Ippel et al. (2024) (the reader is also referred to Calaf et al. (2010), Speakman et al. (2021) and Heck et al. (2023), who implement a similar approach). In brief, in the actuator disk method, the power of a turbine is computed as $P = Tu_d$, where $T$ is the calculated thrust and $u_d$ is the temporally-filtered disk-averaged velocity normal to the disk plane. The thrust depends on the local (or "modified") thrust coefficient $C_T'$ and the disk velocity $u_d$, and is computed as $T = \frac{1}{2}\rho A C_T' u_d^2$, where $\rho$ is the density and $A$ is the rotor area. This implementation provides predictions of power degradation as a function of rotor misalignment (i.e., for yawed or tilted turbines) similar to Speakman et al. (2021) and Heck et al. (2023). The optimisation framework proposed in Sect. 2.1 is nevertheless compatible with any other turbine model, such as the actuator disk with rotation or the actuator line models, which are expected to yield improved turbine wake and power predictions (Lin and Porté-Agel, 2019).

In order to realistically simulate the interaction of the wind farm with the turbulent atmospheric flow, we perform precursor simulations of pressure gradient driven fully-developed neutral atmospheric boundary layers. A free slip condition is applied at the top boundary, and a no-slip condition with a wall model is used at the ground. The precursor atmospheric flow simulations are run until statistical convergence is reached, after which we begin storing two-dimensional planes of the flow field, normal to the streamwise direction, at every timestep. These planes are subsequently used as inflow conditions in the wind farm simulations.

---

[2]https://github.com/xcompact3d

## 3 Layout optimisation (micro-siting)

In layout optimisation, the objective is to maximise the power production of a wind farm with $N$ wind turbines by adjusting their position $\boldsymbol{c} = [\boldsymbol{x}, \boldsymbol{y}]^T$, with $\boldsymbol{x} = (x_1, \ldots, x_N)$ and $\boldsymbol{y} = (y_1, \ldots, y_N)$, within a given space $\boldsymbol{X}$. The space $\boldsymbol{X}$ corresponds to the available land where the wind turbines may be installed. The farm power production is typically a weighted sum of all the different wind states $K$ (wind direction, speed, turbulence) which the farm experiences over a period of time. The turbines are aligned with the mean inflow wind direction in all cases considered in this section (no controllers are used)[3]. The optimisation problem is therefore expressed as

$$\underset{\boldsymbol{c}}{\arg\max} \quad \sum_{k=1}^{K} a_k \sum_{n=1}^{N} P_{n,k}(\boldsymbol{c}) \tag{3a}$$

$$\text{s.t.} \qquad \boldsymbol{c} \in \boldsymbol{X}, \tag{3b}$$

$$\qquad ||\boldsymbol{c}_i - \boldsymbol{c}_j|| > D, \text{ for } i, j = 1, \ldots, N \text{ and } i \neq j \tag{3c}$$

where $P_{n,k}$ is the power output of the $n$-th turbine given wind state $k$, and $a_k$ is the weight (i.e. probability) of each wind state, as dictated by the local meteorological data. Equation (3b) constrains the turbines within the available land. To avoid turbine overlapping, a second constraint is added to ensure that the centres of the turbines are spaced at least one turbine diameter $D$ apart (Eq. (3c)).

In a preliminary proof-of-concept study (Bempedelis and Magri, 2023), an early version of the proposed framework was applied to a simple layout optimisation problem (five wind turbines in a $6D \times 6D$ square, operating under a single wind state). It was shown that the method is capable of finding optimal designs in a few iterations whilst leveraging flow phenomena that are unaccounted for in low-fidelity solvers (e.g., local speed-ups) to deliver increased wind farm performance.

However, modern wind farms are composed of several more turbines. The layout optimisation problem therefore becomes high-dimensional, with the design space being $2N$-dimensional. Bayesian optimisation in low dimensions is well established, but its application to high-dimensional problems poses a significant challenge due to the increase in complexity in both GP regression and the search for the next designs (AF optimisation) (Binois and Wycoff, 2022).

Here, we propose a way to reduce the dimensionality of the layout optimisation problem. It is based on the assumption that, in most cases, the turbines in a wind farm are identical. In that case, the cost function is invariant under permutations of the turbine labelling, and there are $N!$ different combinations of the design variables that produce the same output. Such behaviour is a clear barrier to efficient optimisation. To remedy this, we could augment the training dataset with all $N!$ permutations of the evaluated designs. However, this quickly gives rise to high complexity, as the computational cost for GP regression scales with $\mathcal{O}(m^3)$, where $m$ is the number of points in the training dataset. Alternatively, without loss of generality, we propose working on a reduced design space by enforcing a set of $N-1$ additional constraints (Eq. (4d)). The constraints enforce a particular ordering of the turbines, in one direction only, thereby constraining the design variables to that particular subspace of the full parameter space. The subspace size is smaller by a factor $1/N!$ relative to the original one, indicating the equivalence of the

---

[3]In reality, accurate and robust estimation of the wind direction poses a significant challenge (Annoni et al., 2019).

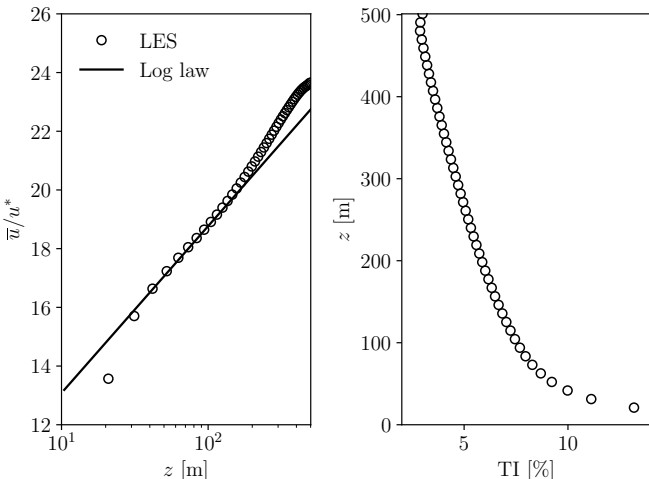

**Figure 1.** Mean streamwise velocity (left) and turbulence intensity (right) of the simulated atmospheric boundary layer.

two approaches. (A third way, not explored in this work, is to encode the turbine ordering invariance property of the objective function in the kernel; see Duvenaud (2014) for additional details). The layout optimisation problem is thus reformulated as

$$\arg \max_{\boldsymbol{c}} \quad \sum_{k=1}^{K} a_k \sum_{n=1}^{N} P_{n,k}(\boldsymbol{c}) \tag{4a}$$

$$\text{s.t.} \quad \boldsymbol{c} \in \boldsymbol{X}, \tag{4b}$$

$$||\boldsymbol{c}_i - \boldsymbol{c}_j|| > D, \text{ for } i,j = 1,\dots,N \text{ and } i \neq j, \tag{4c}$$

$$x_i \geq x_{i-1}, \text{ for } i = 2,\dots,N \tag{4d}$$

### 3.1 Layout optimisation: Problem set-up

We consider a wind farm of $N = 16$ turbines, each with $D = 100$ m rotor diameter and $h = 100$ m hub height. The wind turbines operate with a constant local thrust coefficient $C_T' = 4/3$. The land where the turbine towers can be placed is a square of size $18D \times 18D$ (for this set-up, the blades can extend outside of the available area). We consider an evenly-weighted six-directional wind rose. This is a set-up close to those examined in King et al. (2017) and Antonini et al. (2018). The atmospheric boundary layer is characterised by friction velocity $u^* = 0.442$ ms$^{-1}$, height $\delta = 501$ m and roughness length $z_0 = 0.05$ m, which correspond to conditions in the North Sea (Wu and Porté-Agel, 2015). The velocity at the hub height of the turbines is $U_h = 8.3$ ms$^{-1}$, and the turbulence intensity at the same level is $\text{TI}_h = 7.4\%$ (see Fig. 1 for the simulated atmospheric wind profiles).

The wind farm is embedded in a computational domain of size $4008 \times 3340 \times 501$ m, which is discretised with a uniform mesh with $\Delta x = \Delta y = \Delta z = 10.4375$ m (corresponding to $\approx 10$ points across the turbine rotor). As shown in Bempedelis et al. (2023), this resolution is sufficient to predict the power production of a large wind farm. This was also confirmed for the

considered set-up (result not shown for brevity). The flow field planes stored from the precursor simulation (see Sect. 2.2) are used at the inlet, a convective condition is used at the outlet, and the conditions at the remaining boundaries are similar to the description in Sect. 2.2. The timestep is chosen such that the maximum Courant (CFL) number remains below 0.2. Data are averaged over a 2.5-hour period of farm operation, following the time required for the flow to develop.

Figure 2 shows the contours of the instantaneous and mean streamwise flow at the turbine hub level for a randomly sampled farm layout under westerly wind, together with details on the described set-up (wind rose and available land). It also reveals the highly complex nature of wind farm flows, with turbine wakes interacting with one another and with the atmospheric turbulence.

## 3.2 Layout optimisation: Results

The optimisation starts by evaluating the farm power output of a large set of layouts sampled from a Latin hypercube, along with two user-designed layouts. These are a uniform $4 \times 4$ layout with 6D spacing between the turbines and a layout where the turbines are equidistantly distributed on the land border only. The optimisation then progresses iteratively, as described in Sect. 2, and terminates after 700 iterations (for a total of 4200 wind farm LES). This was decided in consideration of the available computational resources (a comment on the cost of the simulations can be found in Appendix A) and the optimisation's progression. The optimisation history is shown in Fig. 3. To facilitate comparison, we introduce the metric $\eta$, defined as the ratio between the overall farm power output and a reference ideal output based on the output of a single wind turbine

$$\eta = \frac{\sum_{k=1}^{K} a_k \sum_{n=1}^{N} P_{n,k}}{N \times P_{\text{single}}} \tag{5}$$

with $P_{\text{single}}$ being computed in a separate LES, where a single turbine was placed at the center of the available land. Basically, $\eta$ is a normalised form of the objective function defined in Eq. (4a), based on the production of $N$ similar stand-alone turbines. It may therefore be interpreted as a farm efficiency coefficient primarily describing wake losses.

At the end of the optimisation, the best-performing design has $\eta = 95.3\%$ efficiency, with the corresponding mean flow fields shown in Fig. 4 for every wind direction. We observe limited turbine-wake interference, especially in the diagonal wind directions, which is associated with increased power production by the wind farm in those directions. For reference, the $4 \times 4$ aligned layout and the layout shown in Fig. 2 have efficiencies $\eta = 88.5\%$ and $\eta = 80.4\%$, respectively.

In parallel with the training of the surrogate model, we can also learn what constitutes good modelling practices. By means of validation on subsequent test points, we observe that the rational quadratic (RQ) kernel outperforms both the widely used Matérn and the squared exponential (RBF) kernels. Figure 5 shows the predictions of the regressed GPs (with a training dataset of 600 points) for the 40 points tested next. The test points include designs suggested with different levels of balance between exploration and exploitation (by tuning the $\lambda$ parameter in the AF, see Sect. 2.1). All kernels perform adequately in regions near points in the training dataset. However, the RQ kernel is shown to extrapolate more accurately. Similar conclusions hold for both anisotropic (ARD) kernels (in which each design variable is associated with its own lengthscale) and isotropic kernels. In Bempedelis and Magri (2023), it was shown that it is possible to find optimal designs with isotropic kernels. Here, we find that anisotropic kernels show slightly improved accuracy. However, this comes at the price of larger computational cost and

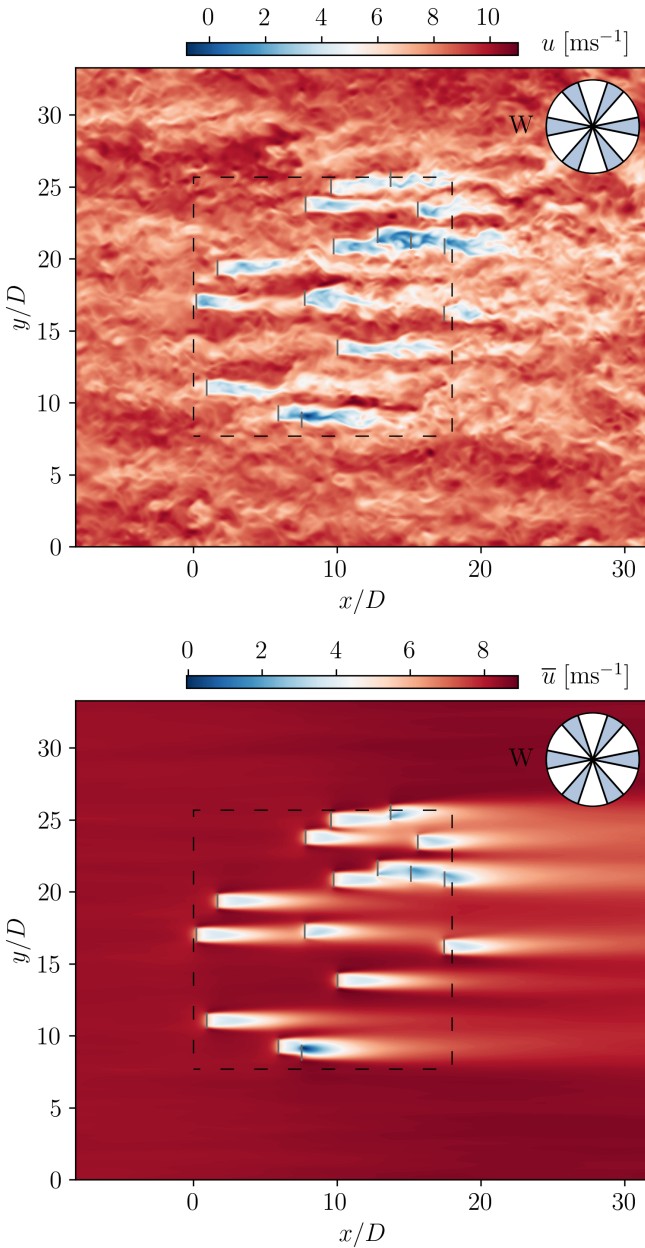

**Figure 2.** (Top) Instantaneous and (bottom) mean streamwise velocity at the turbine hub height for a randomly sampled layout, with the wind blowing from the west. The full six-directional wind rose is shown at the top right. The borders of the available land are indicated with dashed black lines. The wind turbines are denoted with grey solid lines.

complexity. In particular, care needs to be taken when using anisotropic kernels in high-dimensional problems, to ensure that sufficient data are available in the training set to achieve robustness in the GP hyper-parameter optimisation.

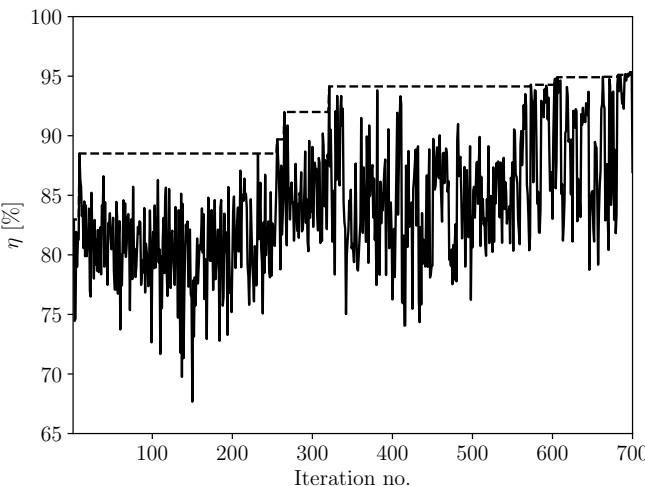

**Figure 3.** Efficiency of designs evaluated during the layout optimisation process. The dashed line shows the evolution of the best-performing design.

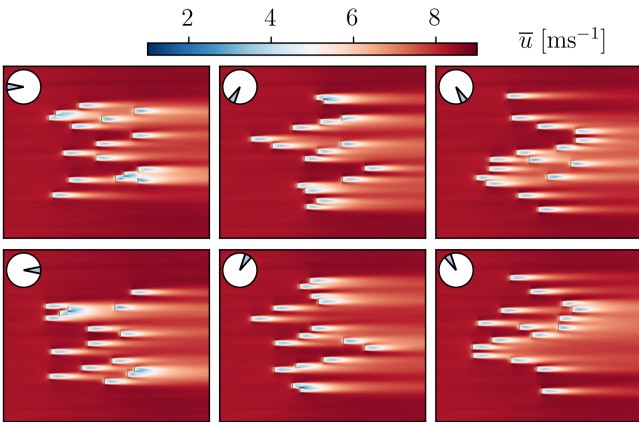

**Figure 4.** Mean flow fields for the best-performing design proposed by the large-eddy simulation Bayesian optimisation framework (LES-BO). The flow fields show the same layout exposed to different wind directions. In all cases, the wind is shown as blowing from left to right, with the wind farms rotated to match the wind direction shown on the inset wind roses. The reference non-rotated layout is that of the westerly wind case (top left).

In order to assess the quality of the best-performing design obtained with the proposed framework (hereafter referred to as LES-BO), we perform a series of optimisations with the FLOw Redirection and Induction in Steady State (`FLORIS`) software (NREL, 2023). `FLORIS` is an open-source wind farm simulator that incorporates several widely used wake models. In this

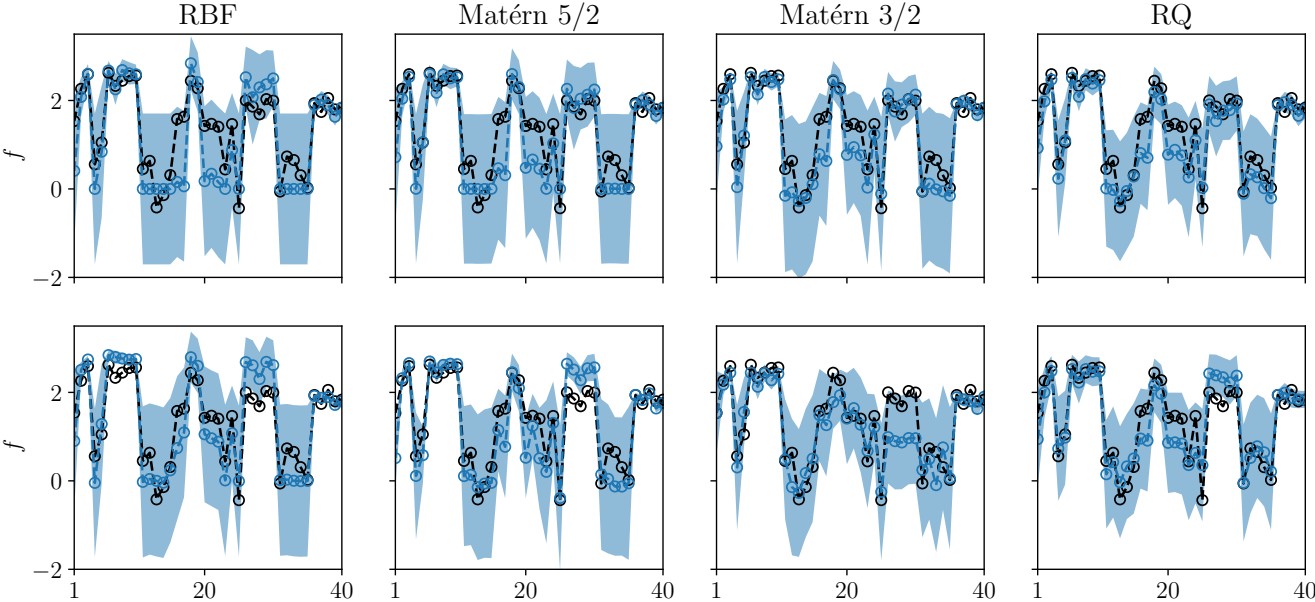

**Figure 5.** GP regression with different covariance functions on a standardised training dataset of 600 points. $f$ is the standardised overall farm power output, $f = (P_F - \mathrm{mean}\,(P_F))/\sigma(P_F)$, with $P_F$ denoting the farm output. Predictions of the regressed models for the 40 points computed next in the optimisation chain. The GP mean predictions are denoted with blue markers. The shaded areas represent 95% confidence intervals. Black markers denote the LES predictions. (Top row): Isotropic kernels. (Bottom row): Anisotropic kernels.

study, the Gauss-Curl Hybrid model (King et al., 2021) is employed for predicting the wind farm flow and power output, while the optimisation algorithm used is the FLORIS-default gradient-based Sequential Least SQuares Programming (SLSQP) method (Kraft, 1988). For more details on the FLORIS simulations, please see Appendix B. In consideration of the multi-
225 modal nature of the layout optimisation problem, and to enable a more comprehensive comparison with the LES-BO method, we perform 100 independent FLORIS layout optimisations, initialised with 99 random layouts and a uniform $4 \times 4$ layout with $6D$ spacing between the turbines.

The efficiencies of the 100 optimal designs outputted by FLORIS are then evaluated using Winc3d (the designs suggested by the low-fidelity model and evaluated with LES will be referred to as LF/LES). The LF/LES results, ordered from best to
230 worst, are presented in Fig. 6, together with the predictions of FLORIS and the best-performing design obtained with LES-BO.

Figure 6 allows us to draw a number of conclusions. First, the proposed LES-BO framework is capable of finding a design that produces more power than $\approx 70\%$ of the optimal designs suggested by FLORIS (as evaluated with LES). Taking into account the relatively small amount of iterations to find this design, the option of using more advanced BO strategies (e.g., Eriksson et al. (2019)), and the fact that our focus is on feasibility rather than performance, we conclude that optimising wind
farms using a high-fidelity surrogate modelling approach is achievable.

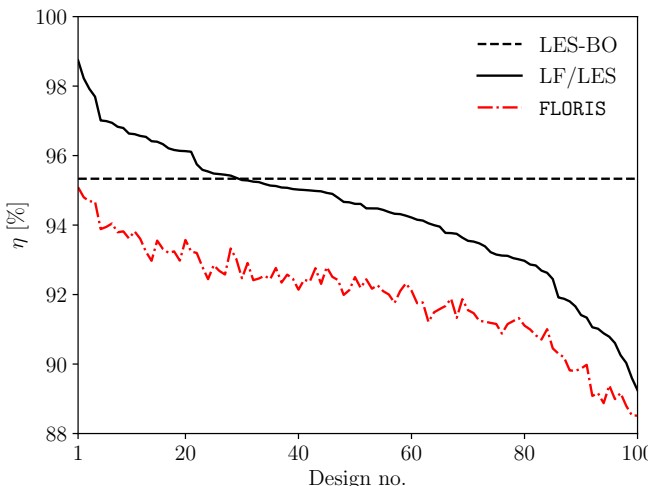

**Figure 6.** Efficiency of optimal designs outputted by `FLORIS`, evaluated with both `FLORIS` and LES (LF/LES). The dashed line shows the optimal design suggested by the LES-BO framework.

However, several layouts suggested by the wake model-based optimisation outperform those suggested by LES-BO. This finding supports the practice of using wake models to design wind farms in the wind energy industry, as they demonstrate excellent performance at only a fraction of the cost (see also Thomas et al. (2019)).

Nevertheless, the efficiency predictions of `FLORIS` are lower compared to those of LES. As shown in Fig. 6, the trends of both predictions are similar, but the offset between the two is not constant, with LES predicting a wider range of efficiencies. Also, it may be observed that there is no one-to-one correspondence between the predictions of the two flow models. These differences are attributed to the flow phenomena that are unaccounted for in `FLORIS`. To identify these, Fig. 7 shows the efficiency of individual turbines of the `FLORIS`-suggested best design, evaluated by both `FLORIS` and LES, for a single wind direction ($270°$) and the full wind rose. In the case of the westerly wind, we observe that the increase in power production comes from downstream turbines. According to LES, the first turbines are producing slightly less power, indicating the negative impact of global (farm-level) blockage that is not taken into account in the wake models. However, the production of downstream turbines is increased, to levels above those of an isolated turbine, due to local flow speed-ups. Here, an additional simulation with three times larger spanwise extent was performed to evaluate the effects of domain blockage. These were confirmed to be present but relatively small, with $\approx 0.43\%$ difference in maximum streamwise velocity at hub height and $\approx 1.36\%$ difference in farm power production. Similar benefits owing to local blockage were also reported in King et al. (2017) and Antonini et al. (2018). Nevertheless, it is important to highlight that blockage effects are particularly sensitive to the atmospheric conditions besides the extent of the computational domain (Bleeg and Montavon, 2022). In the case of the full wind rose, all turbines operate at an efficiency above $90\%$, with some turbines exceeding, on average, the production of an isolated turbine. Overall, the individual turbine efficiency predictions of the two flow models for the full wind rose (aggregate power) are qualitatively similar, with varying levels of differences.

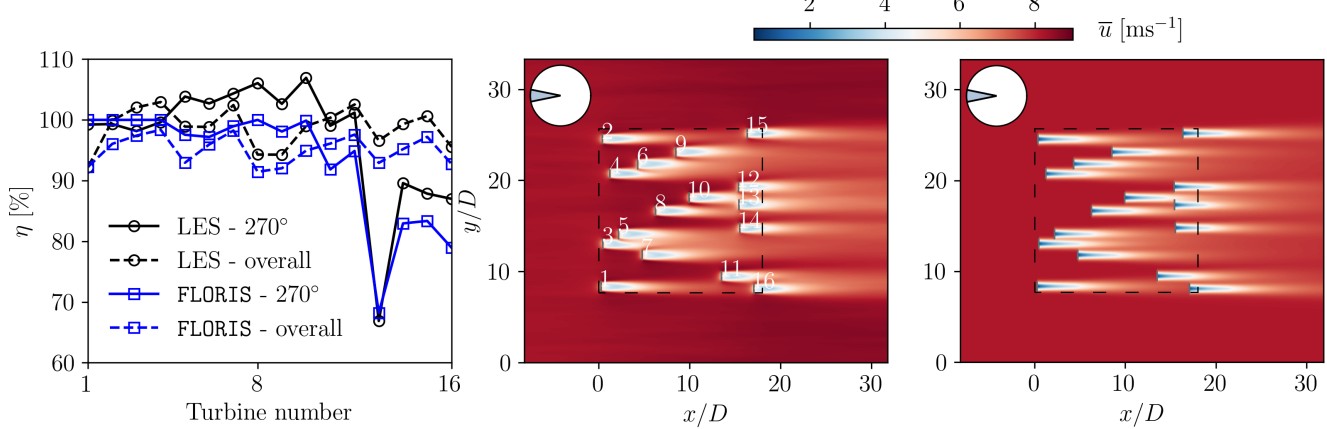

**Figure 7.** `FLORIS`-suggested best design. (Left): Individual turbine efficiencies predicted by `FLORIS` and LES. (Middle) Mean streamwise flow for westerly wind, LES. (Right) Mean streamwise flow for westerly wind, `FLORIS`.

Figure 7 also shows the mean streamwise flow fields predicted by `FLORIS` and LES for the westerly wind case. Together with Fig. 8, which shows the LES predictions for the in-plane velocity angle and the turbulence intensity, we observe several phenomena, in addition to the local speed-ups, which are taken into account only in LES (see also figures 2 and 4). These include cross-stream flow, wake deflections and curvature (e.g., at the sides of the farm), pressure gradients, and wake merging (e.g., turbines 11 and 16). Another difference between the predictions of the two flow models is in the near wake of the turbines. Wake models assume that self-similarity is established immediately downstream of a turbine, which is known to be inaccurate (see e.g., Steiros et al. (2024)). Although this may not be a significant concern in the relatively sparse arrangement considered in this work, it is expected to play a more significant role in farms with higher turbine density. Finally, Fig. 9 shows a smoke-type visualisation of the turbine wakes for the `FLORIS`-suggested best farm layout. To achieve this, a passive scalar is emitted from the turbine rotors and is transported by the flow. This enables us to observe the meandering of the wakes, and possibly some indications of weakly—owing to the stabilisation effects of base bleed (Steiros et al., 2020, 2021)—coupled dynamics (e.g., turbines 12,13 and 14) (Peschard and Le Gal, 1996).

### 3.3 Combining wake models with LES: A multi-fidelity approach

Section 3.2 demonstrated the feasibility of optimising the layout of wind farms with the proposed LES-BO framework, along with the role of the mechanisms that are neglected by low-fidelity flow solvers. However, for the number of optimisation iterations performed, LES-BO was outperformed by the multi-start gradient-based optimisation with low-fidelity models.

In order to show the benefits of adopting the LES-BO approach, we propose a simple multi-fidelity strategy. We train a surrogate model with a dataset composed of only the first 300 samples from our original investigation (see Fig. 3) and the 100 designs suggested by the low-order model (see Fig. 6). Following initial training, the optimisation proceeds as discussed

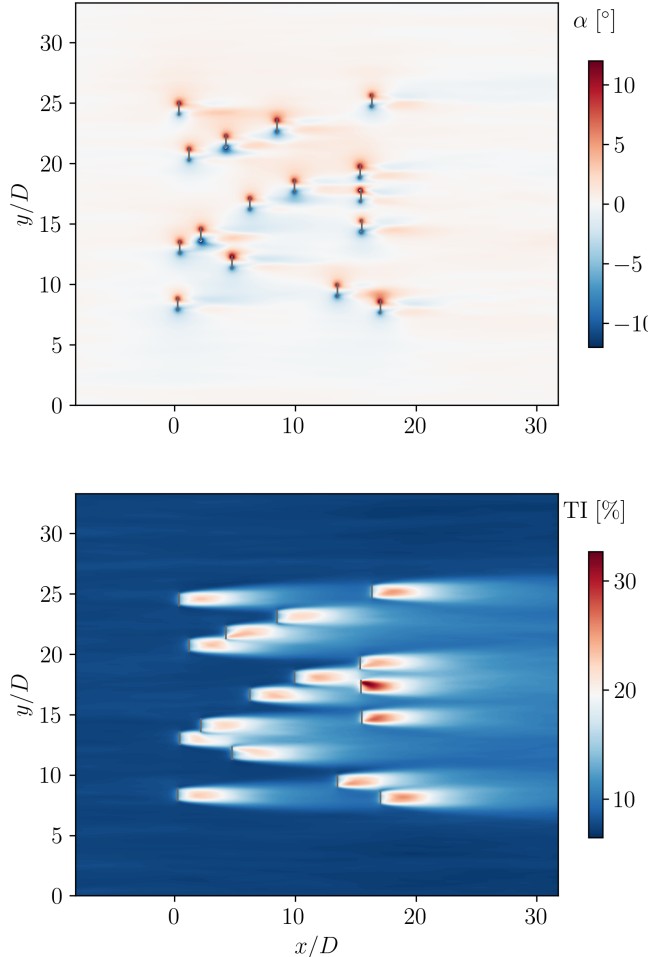

**Figure 8.** `FLORIS`-suggested best design. LES predictions for (top) angle of horizontal plane velocity vector, (bottom) turbulence intensity.

in Sect. 2, and similar to Sect. 3.2. However, our surrogate is now informed with well-performing designs obtained from the low-fidelity optimisation. We refer to this strategy as the extrema-informed LES-BO framework (EI-LES-BO).

Within just thirty iterations, EI-LES-BO manages to improve upon the `FLORIS`-suggested best design by $\delta\eta = 0.32\%$. This is achieved by leveraging the flow phenomena that are unaccounted for in `FLORIS`. Figure 10 shows the efficiency of the two designs for each direction in the wind rose. EI-LES-BO delivers increased performance for most wind directions.

## 4  Wake steering (yaw angle optimisation)

In wake steering, the objective is to maximise the power output of a wind farm by adjusting the angle $\gamma$ at which turbines face the wind (a yaw angle $\gamma = 0°$ corresponds to the turbines being aligned with the mean wind direction). The position and

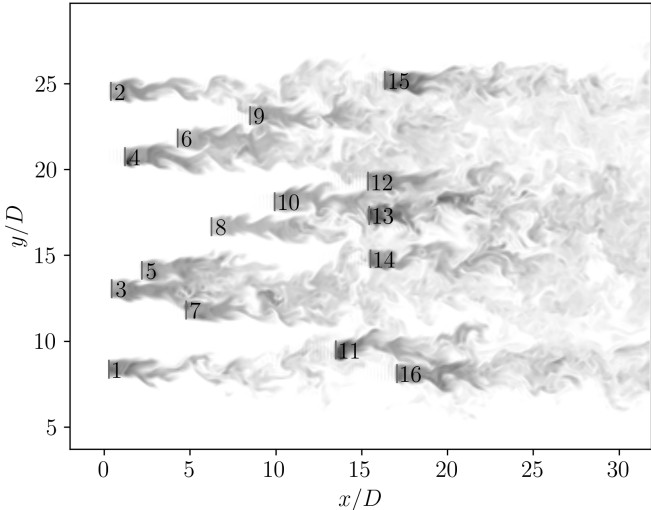

**Figure 9.** Wake visualisation in the `FLORIS`-suggested best design case by means of transport of a passive scalar shed from the turbine rotors.

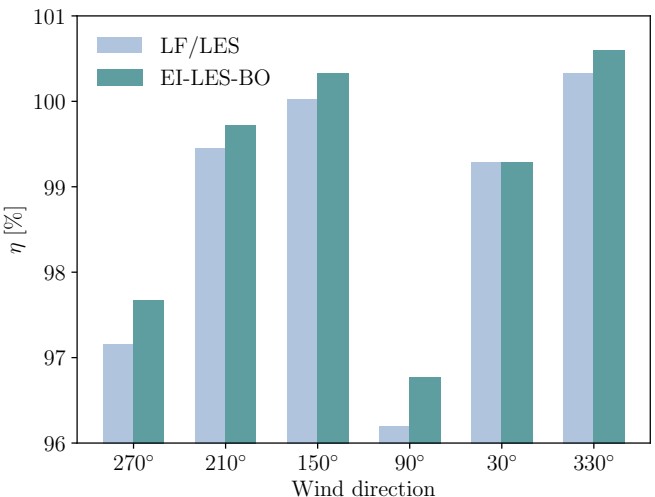

**Figure 10.** Efficiency of best-performing LF/LES and EI-LES-BO layouts for each direction in the wind rose.

properties of the wind turbines are considered known. As a result, optimisation can be carried out independently for each wind state. The (static) yaw angle optimisation problem for a single wind state can then be expressed as

$$\arg\max_{\boldsymbol{\gamma}} \quad \sum_{n=1}^{N} P_n(\boldsymbol{\gamma}) \tag{6a}$$

$$\text{s.t.} \quad \boldsymbol{\gamma} \in \Gamma \tag{6b}$$

where $\boldsymbol{\gamma} = (\gamma_1, \ldots, \gamma_N)$ denotes the turbine yaw angles and $\Gamma$ is the range of admissible misalignment angles.

## 4.1 Wake steering optimisation: Problem set-up

The problem we consider draws on the Horns Rev I wind farm, which is located in the North Sea, and consists of eighty turbines with diameter $D = 80$ m and hub height $h = 70$ m, arranged in an $8 \times 10$ grid with $7D$ spacing between the turbines (Barthelmie et al., 2009). Data for the atmospheric boundary layer are taken from Wu and Porté-Agel (2015), with $u^* = 0.442$ ms$^{-1}$, height $\delta = 500$ m and roughness length $z_0 = 0.05$ m (similar to the conditions considered in Sect. 3). Under these conditions, the velocity and turbulence intensity at hub height are $U_h = 7.9$ ms$^{-1}$ and TI$_h = 8.0\%$, respectively.

We assume that the turbines operate with a constant local thrust coefficient, $C_T' = 4/3$, and may yaw up to $\pm 30°$ ($\Gamma \in [-30, 30]$). We consider a single wind direction ($270°$), which is among the most prevalent ones, and the wind is aligned with the wind farm rows. This case is selected because wake steering optimisation studies based on low-fidelity models have yielded wildly varying optimum yaw settings, which range from zero to the maximum admissible yaw angle (see e.g., Zong and Porté-Agel (2021); Gori et al. (2023)). For the purposes of the present optimisation study, we simulate only one row of the wind farm and use periodic conditions on the lateral boundaries. This follows a number of studies that have reported row independence for the considered problem (Zong and Porté-Agel, 2021; Gori et al., 2023). For instance, Zong and Porté-Agel (2021) reported an estimated asymptotic wake deflection of $0.8D$ under a yaw condition of $\gamma = 30°$, effectively demonstrating that row-to-row interactions have negligible influence on wake steering effectiveness for the investigated turbine spacings.

The domain of size $6144 \times 560 \times 1024$ m is discretised with a uniform mesh with $\Delta x = \Delta y = \Delta z = 8$ m (corresponding to 10 points across the turbine rotor, a resolution equivalent to the one used in Sect. 3). Data are gathered over a 3-hour period of farm operation, following the time required for the flow to develop. Figure 11 shows contours of the instantaneous and mean streamwise flow at the turbine hub level, for the case of non-yawed conditions, ($\boldsymbol{\gamma} = 0°$).

## 4.2 Wake steering optimisation: Results

The initial training dataset comprises fifty designs. These include yaw angle combinations sampled with the Latin Hypercube method, the non-yawed condition shown in Fig. 11, and a design in which the yaw angles decrease linearly, from the maximum admissible yaw angle for the most upstream turbine to zero yaw angle for the most downstream one (Zong and Porté-Agel, 2021). Significant improvements in power output are observed within just ten iterations of the LES-BO optimisation. To facilitate comparison between all considered conditions, we define the metric $\eta$ as the ratio of improvement over the non-yawed case, namely when $\boldsymbol{\gamma} = 0°$

$$\eta = \frac{\sum_{n=1}^{N} P_n(\boldsymbol{\gamma})}{\sum_{n=1}^{N} P_n(\boldsymbol{\gamma} = 0°)} \tag{7}$$

The above metric is a measure of power output improvements when adopting wake steering. Similar to Sect. 3, the best performing LES-BO design is compared with the `FLORIS`-suggested optimal design (Fig. 12). In particular, Fig. 12 shows the optimal yaw angles suggested by each framework, along with the corresponding LES predictions for individual turbine efficiency. The latter is defined as $\eta_{\text{ind},i} = P_i(\boldsymbol{\gamma})/P_i(\boldsymbol{\gamma} = 0°)$.

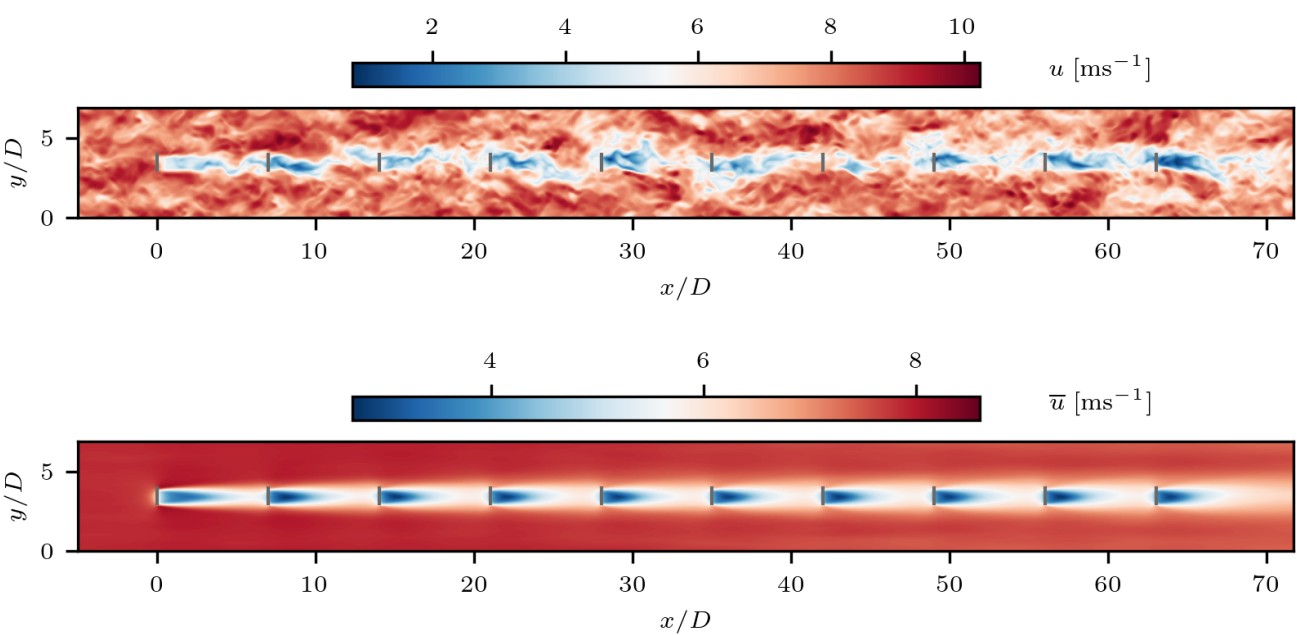

**Figure 11.** (Top) Instantaneous and (bottom) mean streamwise velocity at the turbine hub height for non-yawed conditions.

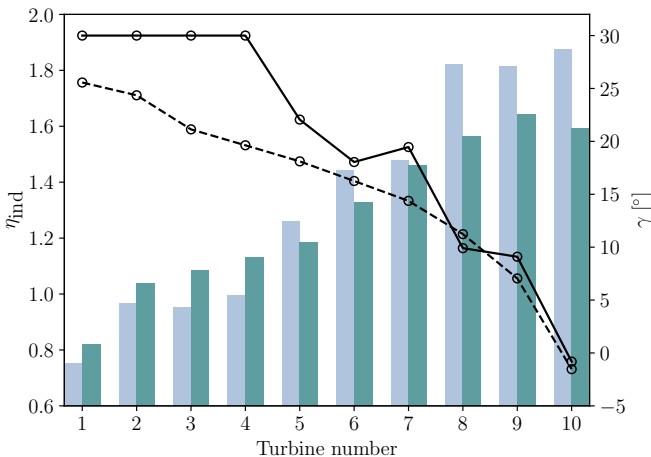

**Figure 12.** Optimal yaw angles suggested by LES-BO (solid line) and `FLORIS` (dashed line). Individual turbine efficiencies are shown as bars with solid fill (LES-BO) and bars with hatched pattern (LF/LES).

Qualitatively, both frameworks suggest a decreasing trend in yaw angles as we move downstream in the farm, similar to results reported in Bastankhah and Porté-Agel (2019). However, differences in magnitude can be observed, with the LES-BO framework favouring larger yaw angles in the upstream turbines. This results in reduced power output for the first four turbines.

For downstream turbines, both frameworks suggest yaw angles of similar magnitudes. However, because they are less affected by the wakes of upstream turbines, downstream turbines in the LES-BO design produce more power.

As actuator disk theory tends to overestimate the power output of wind turbines at large yaw angles (Lin and Porté-Agel, 2019), both frameworks are prone to over-promoting wake steering. Nevertheless, the operational details of the turbines are identical in both flow models (LES-ADM and FLORIS). This suggests that the preference for larger yaw angles in the LES is likely related to the flow mechanisms (e.g., row interactions) that are unaccounted in FLORIS. We note that the yaw angles of the first four turbines in the LES-BO design are at the arbitrarily selected yaw limits defined in the problem set-up (see Sect. 4.1). This means that there is potential for further improvements should these limits be extended or removed. In practice however, for large yaw offsets, it is important to also consider the effects of yawing the turbine on the loads it experiences (Damiani et al., 2018; Ennis et al., 2018).

Figure 13 shows the mean streamwise velocity field for the two cases (LES-BO best and LF/LES). Overall, the LES-BO framework is able to find a design that is $4\%$ more efficient compared to the design suggested by FLORIS, with efficiencies $\eta = 1.28$ and $\eta = 1.24$, respectively. $0.7\%$ out of the $4\%$ increase in efficiency is provided by leveraging the effects of confinement (blockage). This is measured by testing the designs in LES with five times the spanwise extent of the domain, effectively simulating them as individual rows of turbines. The difference in efficiency between the designs in the unconfined case reduced to $3.3\%$. Besides affecting the flow both locally and globally (see discussion in Sect. 3.2), confinement also affects wake recovery through increased levels of shear.

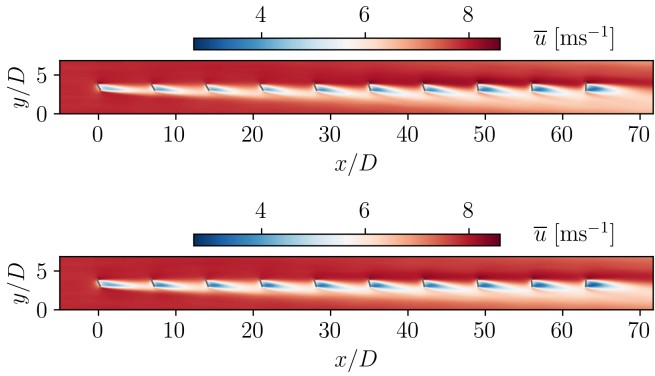

**Figure 13.** Mean streamwise flow at the turbine hub height. (Top) LES-BO, (bottom) LF/LES.

## 5 Conclusions

This work introduces an optimisation framework aimed at enhancing the efficiency of power production in wind farms. The proposed method follows a Bayesian approach and utilises surrogate models based on high-fidelity large-eddy simulations of wind farm flows. As part of an extensive computational campaign involving around 5000 large-eddy simulations, the framework

was effectively used to mitigate losses caused by wake effects through two distinct strategies: layout optimisation (also known as micro-siting), and wake steering through yaw angle optimisation. The achieved optimisation outcomes were also compared with results obtained from low-fidelity wake model-based optimisation. Finally, a simple strategy was proposed to combine both large-edddy simulations and wake models in a multi-fidelity approach.

In the layout optimisation problem, the best layouts found had $\approx 10\%$ increased efficiency compared with the efficiency of a standard aligned layout. The performance of wake models was also found to be remarkable. This is particularly important given the excessive computational requirements of LES-based layout optimisation. In the case of wake steering, the framework relying on high-fidelity simulations outperformed the wake model-based optimisation by a considerable margin ($4\%$ increased efficiency, reaching an overall $28\%$ increase compared with the non-yawed conditions). The increased complexity of the flow in this scenario rendered the utilisation of high-fidelity flow models advantageous. In future work, this framework will be extended to other flow cases of increased complexity, such as wind farms on complex terrain or under stratified conditions.

## Appendix A: Computational cost

Simulations were performed using $128$ or $256$ CPU cores on ARCHER2 (https://www.archer2.ac.uk/) and CSD3 (https://www. csd3.cam.ac.uk/). The cost of each simulation ranged from $\approx 300$ to $\approx 900$ CPUh, mostly depending on the type and amount of outputted data. These included instantaneous and time-averaged fields, and time-resolved probe and turbine data.

## Appendix B: Low-order modelling and optimisation

Low-order farm modelling is conducted using version 3.4 of `FLORIS` (NREL, 2023), an open-source wind farm simulator developed by the National Renewable Energy Laboratory (NREL). The framework incorporates several widely-used steady-state analytical wake models to predict the wind farm flow and power output. In this study, we employ the Gauss-Curl Hybrid wake model (King et al., 2021), with the deflection model of Bastankhah and Porté-Agel (2016), and the sum of squares freestream superposition (SOSFS) model of Katic et al. (1987). For all models, we use the `FLORIS` default parameters. The atmospheric conditions specified in sections 3.1 and 4.1 are matched by specifying a wind profile with a power law relationship (with a best fitting exponent $0.133$) and turbulence intensity at the hub height as computed in the LES.

In terms of optimisation, the algorithm of choice is the FLORIS-default gradient-based Sequential Least SQuares Programming (SLSQP) method (Kraft, 1988). In the farm layout optimisation, 100 different optimisations are independently conducted, each starting from different initial conditions. The SLSQP parameters $ftol = 10^{-9}$ and $eps = 0.01$ are set following `FLORIS` recommendations (NREL, 2023). In the wake steering optimisation problem, the optimal yaw angles obtained in Gori et al. (2023) are set as initial conditions to avoid initialisation sensitivity issues. Furthermore, the SLSQP parameters $ftol = 10^{-12}$ and $eps = 0.05$ are defined as suggested in Gori et al. (2023) for wake steering application on the Horns Rev wind farm.

*Data availability.* The data that support this study are available upon reasonable request.

*Acknowledgements.* The authors would like to thank Mr. Christian Jané-Ippel for his work in the development of the flow solver. NB, SL and LM are supported by EPSRC, Grant No. EP/W026686/1. SL and LM also acknowledge support from EPSRC, Grant No. EP/Y005619/1. LM also acknowledges support from the ERC Starting Grant PhyCo No. 949388 and the EU-PNRR YoungResearcher TWIN ERC-PI_0000005. The authors would like to thank the UK Turbulence Consortium (EP/R029326/1 and EP/X035484/1) for providing access to ARCHER2. We also acknowledge use of the Cambridge Service for Data Driven Discovery (CSD3) system. FG would like to acknowledge the Department of Aeronautics, Imperial College London, for supporting this work with a doctoral studentship.

*Author contributions.* Nikolaos Bempedelis: Conceptualisation, Methodology, Investigation, Writing - Original Draft. Filippo Gori: Investigation, Writing - Review & Editing. Andrew Wynn: Writing - Review & Editing. Sylvain Laizet: Writing - Review & Editing, Funding acquisition. Luca Magri: Conceptualisation, Writing - Review & Editing, Funding acquisition.

*Competing interests.* The authors declare that they have no conflict of interest.

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
