# Peer review of "Data-driven optimisation of wind farm layout and wake steering with large-eddy simulations"

_Wind Energy Science, 2023_

## Referee Comment (RC1)

Review of:
"Data-driven optimisation of wind farm layout and wake steering with large-eddy simulations"
by Nikolaos Bempedelis, Filippo Gori, Andrew Wynn, Sylvain Laizet, and Luca Magri

**General Comments:**

The article present two different studies on optimization of wind farm performance, one on layout optimization and one on wind farm control using wake steering. The optimizations are performed using high-fidelity (LES) and low fidelity (FLORIS) wake modelling as well as multi-fidelity by combining results from both. The results are based on an impressive amount of LES, and provide interesting results. However, the article also show several shortcomings and several aspects, which lack important details for instance in terms of modeling choices, numerical blockage and uncertainties. Therefore, I recommend major revisions.

**Specific Comments:**

**Lack of details:**

**Actuator Disc modeling and Power estimation:**

The description of the actuator disc method is very inadequate, and Bempedelis et al. 2023 does not provide the necessary details. I can only assume that an uniformly loaded actuator disc is employed. If so, it is far from state-of-the-art and it has been shown several times to be insufficient, particular for more complex flows such as yawing turbines, see for instance [1], where a standard (uniform) actuator disc is compared to a BEM based actuator disc with rotation.

[1] also show how power production behind a uniformly loaded actuator disc is significantly overestimated compared to a slightly more advanced actuator disc model using BEM and rotation. Therefore, the authors should specify how the power is estimated, as this is particular important for yawing turbines. BEM has been shown to not give good estimates of power production in yaw, and this is further complicated if yawing turbines are operating in waked conditions, where power production can be overestimated, see [2]. I suspect the efficiencies in Figure 12 are a result of simplification, but it is hard to test/decipher as a reader. It would be beneficial to also report the actual power production (in [kW]), not only the efficiency. Furthermore, I have some concerns about the optimization angles shown in Figure 12. First, it is a general concern when optimizations consistently provides results at the admissible limit, as is the case where the first four turbines are yawing 30°. Second, several higher fidelity simulations have shown that there are secondary effects of wake steering. This means the turbines operating in wake will typically yaw less, see [3] for two turbines, where the second turbine yaws positively to increase production, and [4], where deep-row turbines should yaw relatively less. Such secondary effects are actually also visible in Figure 8.

**Numerical blockage:**

For the second study, the authors assess that the estimated efficiency would reduce from 4% to 3.3% by reducing the numerical blockage from $(\pi \cdot 80m^2)/(560m \cdot 1024m) = 3.5\%$ to $(\pi \cdot 80m^2)/(5 \cdot 560m \cdot 1024m) = 0.7\%$ (I assume it is a typo that line 282 states km, not m). This analysis is good and the results plausible. However, in the first study, the numerical blockage is up to $(16 \cdot \pi \cdot 100m^2)/(3340m \cdot 501m) = 30\%$ and even if the 16 turbines were arranged in a $4 \times 4$ it would correspond to 7.5%. For such layouts, the numerical blockage is significant and the impact clearly seen in the results with substantial speedups, where certain turbines have efficiencies of 110% (Figure 7). I am sceptical how realistic these speed-ups are, particular given the sensitivity mentioned for the second study. The article states that the impact is less than 1% when tested with a domain three times wider (line 230-231) and refers to Antonini et al. (2018) and King et al. (2017). I can not find the details in Antonini et al., which is also a 2D simulation, but King et al. have blockage ratio of 5.2% for worst case scenario and reference Chen and Liou to give a threshold of less than 10% for wind tunnel studies. I think the details of this analysis is required in an appendix.

**Uncertainties:**

- line 126: In principle, turbines should align themselves with the incoming wind direction. The reality is however, that unintended yaw misalignment is a very large uncertainty in normal operating wind farms, and it is a large challenge in order to actually apply wind farm control, where it is notoriously difficult to determine a wind direction and hence provide accurate estimates of how much to yaw, see for instance [5, 6].
- Section 2.1 and line 206: SLSQP is used to optimize FLORIS while LES is optimized with BO, but does the choice of optimization strategy not potentially affect the optimization results? Please motivate why different methods are used.
- There are significant uncertainties of wake steering related to the impact on structural loads, particular for large yaw angles, which is not addressed.

**Unclear comments:**

- Line 21: I think it is misleading to say that wind farms become increasingly less efficient as their size increase. First, many studies have shown how there an equilibrium between power extracted by turbines and the entrained energy from the atmosphere, see e.g. [7]. Second, wind farm layouts have historically developed from aligned/rectangular layouts to curved and finally more misaligned and "random" layouts. The authors use Horns Rev for the second study, and this development in layouts is very clear in the three different generations of wind farms at Horns Rev, see figure 1. The figure is reproduced from [8], where it can also be seen (Table 2) how the capacity factor has increased historically for these wind farms.
- line 34: Wake steering is still not generally applied on commercial wind farms, despite growing scientific evidence. The main reason is that the uncertainties related to wake steering remains very high, i.e. it is not given that overall power output will increase, see for instance the review paper [9] or the benchmark paper [10]. I think the article would benefit for including such considerations at least in the motivation and discussion.
- line 55: RANS is fully capable of modeling wake-to-wake interaction.
- Figure 6 and text: I find the text unclear related to the figure. First, in line 215 it says that 70% of the designs found by LES-BO produce more power than the designs found by FLORIS. Looking at Figure 6, LES-BO (black line) is alway above FLORIS (red line), i.e. 100%. The 70% seems to be that the combination of LF/LES outperforms LES-BO. Line 219 is also unclear as to which wake model outperforms LES-BO, but it appears to be LF/LES. It is not clear in the figure, but

have FLORIS and LF/LES been sorted independently or does design "n" of FLORIS correspond to design "n" of LF/LES? If not, it would be interesting to see a correlation plot of the two.

- Section 4: The two studies are somewhat disjointed, and could perhaps even have been clearer in two independent articles. Why not optimize for yaw angle on the optimal wind farm layout of Section 3?

[Figure]

Figure 1: Layout of three generations of wind farms at Horns Rev.

**Minor Comments and Technical Corrections:**

- Figure 5: Please define "f" (y-axis)
- Figure 9: Please add turbine numbers.
- Figure 10: What does the hatched bars indicate?

**References**

[1] Lin M and Porté-Agel F 2019 *Energies* **12** ISSN 1996-1073

[2] Liew J, Urbán A M and Andersen S J 2020 *Wind Energy Science* **5** 427–437 URL `https://wes.copernicus.org/articles/5/427/2020/`

[3] Hulsman P, Andersen S J and Göçmen T 2020 *Wind Energy Science* **5** 309–329

[4] Archer C L and Vasel-Be-Hagh A 2019 *Sustainable Energy Technologies and Assessments* **33** 34–43 ISSN 2213-1388

[5] Quick J, Annoni J, King R, Dykes K, Fleming P and Ning A 2017 *Journal of Physics: Conference Series* **854** 012036 URL `https://dx.doi.org/10.1088/1742-6596/854/1/012036`

[6] Annoni J, Bay C, Johnson K, Dall'Anese E, Quon E, Kemper T and Fleming P 2019 *Wind Energy Science* **4** 355–368 URL `https://wes.copernicus.org/articles/4/355/2019/`

[7] Calaf M, Meneveau C and Meyers J 2010 *Phys. Fluids* **22**

[8] Sørensen J N and Larsen G C 2021 *Energies* **14** ISSN 1996-1073 URL `https://www.mdpi.com/1996-1073/14/2/448`

[9] Kheirabadi A C and Nagamune R 2019 *J. Wind Engin. Ind. Aero.* **192** 45–73

[10] Göçmen T, Campagnolo F, Duc T, Eguinoa I, Andersen S J, Petrović V, Imširović L, Braunbehrens R, Liew J, Baungaard M, van der Laan M P, Qian G, Aparicio-Sanchez M, González-Lope R, Dighe V V, Becker M, van den Broek M J, van Wingerden J W, Stock A, Cole M, Ruisi R, Bossanyi E, Requate N, Strnad S, Schmidt J, Vollmer L, Sood I and Meyers J 2022 *Wind Energy Science* **7** 1791–1825 URL `https://wes.copernicus.org/articles/7/1791/2022/`

---

## Referee Comment (RC2)

**Review: *Data-driven optimisation of wind farm layout and wake steering with large-eddy simulations***

**Summary**

The authors describe an optimization method that uses data from low- and high-fidelity models employing a Bayesian framework. The authors test their optimization method in maximizing wind farm power production through micro-siting and wake steering. The authors compare the best-performing wind farm layout design from their optimization model against optimized wind farm layouts using FLORIS. The proposed framework can generate layouts with similar wind farm efficiency when compared to the optimized layouts obtained using FLORIS. For wake steering, the proposed optimization framework can overperform as compared to optimization using FLORIS. The LES-informed framework can leverage the high-fidelity model capabilities in capturing complex flow features for wind turbine siting and wake steering. The manuscript is well written, and the results are very interesting. However, I recommend major revisions to incorporate important details in the methodology and results.

**Major comments:**

1. Incomplete description of LES framework: The authors perform an impressive number of large-eddy simulations, but the description of the model setup is lacking. The authors are simulating atmospheric flow, but do they incorporate Coriolis in their simulations? Is there a capping inversion in their model, or is the potential temperature profile constant over the entire domain? What are the boundary conditions for the LES used for wind farm layout optimization (Monin-Obukhov similarity at the surface? periodic lateral BC?)? The actuator disk model uses a constant thrust coefficient (not realistic), but how is turbine power estimated (especially for partially waked conditions, like in Figure 2)? The turbine's thrust coefficient changes with yaw angle (Gebraad et al., 2017), which might partially explain the extreme yaw misalignment for the first three turbines in Section 4.

2. Blockage and speedups: The authors report that front-row turbines produce less power than a stand-alone turbine due to blockage, and that downstream turbines can produce more power than a stand-alone turbine due to speedups. I think these statements need to be explained further. Bleeg and Montavon (2022) show the importance of including a capping inversion in the simulation domain and the sensitivity to domain size for simulating blockage. Regarding speedups, the maximum wind speed in Figure 4 appears to be close to 9 m s$^{-1}$, which is an ~8% speedup compared to freestream conditions. Furthermore, some downstream turbines are producing ~10% more power than a stand-alone turbine. These speedups can be an artifact of the width of the numerical domain. How did these speedup regions change when you tested the 3 times wider numerical domain?

3. Computational requirements of this approach: The authors compare the optimized layouts obtained from LES- and FLORIS-informed frameworks, showing that the LES can

produce better results about 70% of the time. It is important to highlight the computational requirements needed to perform the LES- and FLORIS-informed optimizations given that the layouts from FLORIS can overperform when compared to the LES-BO methodology. Furthermore, how realistic is performing 4200 LES for wind turbine siting as compared to optimizing the layout using FLORIS and then evaluating multiple possible layouts using LES?

4. The authors show the capability of their methodology for optimizing a wind farm's layout and wake steering for a single turbine row. Can these two problems be addressed in the same optimization problem? Also, how feasible is it to optimize the yaw angles for wake steering for a whole wind farm rather than for a single turbine row?

**Minor comments:**
1. Figure 4: Rotating the reference frame in Figure 4 can be confusing for the reader. It might seem as if multiple layouts are being tested rather than a single layout for multiple wind directions.
2. What are the intermittent vertical lines in Figure 9 that appear in front of some turbines (e.g., turbines 11, 12, 13, 14, 16)?
3. FLORIS can incorporate varying thrust coefficients for waked turbines. Did you try incorporating a thrust curve in your actuator disk model so that the velocity deficit in waked turbines is not underestimated?

---

## Author Comment (AC1)

**Reply to reviewer #1**
**Manuscript wes-2023-110**

**Nikolaos Bempedelis, Filippo Gori, Andrew Wynn, Sylvain Laizet, and Luca Magri**

The remarks of the reviewer are repeated in blue point by point followed by our answers, while the main modifications in the paper are red-marked in the revised manuscript.

General Comments:
The article present two different studies on optimization of wind farm performance, one on layout optimization and one on wind farm control using wake steering. The optimizations are performed using high-fidelity (LES) and low fidelity (FLORIS) wake modelling as well as multi-fidelity by combining results from both. The results are based on an impressive amount of LES, and provide interesting results. However, the article also show several shortcomings and several aspects, which lack important details for instance in terms of modeling choices, numerical blockage and uncertainties. Therefore, I recommend major revisions.

We thank the reviewer for the time devoted to reading our manuscript and for providing helpful comments and feedback.

Specific comments:
Actuator Disc modeling and Power estimation:
The description of the actuator disc method is very inadequate, and Bempedelis et al. 2023 does not provide the necessary details. I can only assume that an uniformly loaded actuator disc is employed. If so, it is far from state-of-the-art and it has been shown several times to be insufficient, particular for more complex flows such as yawing turbines, see for instance [1], where a standard (uniform) actuator disc is compared to a BEM based actuator disc with rotation.
[1] also show how power production behind a uniformly loaded actuator disc is significantly overestimated compared to a slightly more advanced actuator disc model using BEM and rotation. Therefore, the authors should specify how the power is estimated, as this is particular important for yawing turbines. BEM has been shown to not give good estimates of power production in yaw, and this is further complicated if yawing turbines are operating in waked conditions, where power production can be overestimated, see [2]. I suspect the efficiencies in Figure 12 are a result of simplification, but it is hard to test/decipher as a reader. It would be beneficial to also report the actual power production (in [kW]), not only the efficiency. Furthermore, I have some concerns about the optimization angles shown in Figure 12. First, it is a general concern when optimizations consistently provides results at the admissible limit, as is the case where the first four turbines are yawing 30°. Second, several higher fidelity simulations have shown that there are secondary effects of wake steering. This means the turbines operating in wake will typically yaw less, see [3] for two turbines, where the second turbine yaws positively to increase production, and [4], where deep-row turbines should yaw relatively less. Such secondary effects are actually also visible in Figure 8.

We thank the reviewer for the thorough analysis and comment. The wind turbines are modelled with a standard non-rotational actuator disk model. In section 2.2 of the revised manuscript, we have added a brief description of how power is computed in the actuator disk method, along with four more references [3, 10, 7, 9] besides [2]. Additionally, we have added a link that directs

the interested reader to the open-source code repository for direct access to all methodology and implementation details (`https://github.com/xcompact3d`).

As suggested by the reviewer, the ADM with rotation is a more advanced turbine representation model compared to the standard ADM that we employ in the present study. Nevertheless, the optimisation framework we propose is compatible with any type of turbine representation model (e.g., standard actuator disk, actuator disk with rotation, or actuator line). This is now mentioned in section 2.2 of the revised manuscript, together with a comment on the potential for improved turbine wake and power predictions through use of more advanced turbine modelling approaches.

The efficiencies shown in figure 12 are calculated as $\eta_{ind,i} = P_i(\boldsymbol{\gamma})/P_i(\boldsymbol{\gamma} = 0°)$ (see section 4.2 in the manuscript). Efficiencies are used as a more suitable metric for the comparisons made in this study, as opposed to presenting dimensional power values. The actuator disk method inherently presumes a connection between thrust and power coefficients [10] unless the user specifies otherwise [12]. In our study, where constant thrust (and power) coefficients are assumed, employing efficiency metrics effectively conveys the trends without the need to specify a power coefficient that is representative of a specific turbine only.

Secondary steering effects are accounted for in our simulations. LES-ADM naturally captures the counter-rotating vortices that are present in the wake of a yawed turbine and lead to its curled shape (see, for example, [8]). This is evidenced by figures 12 and 13, where the wakes of downstream turbines deflect by approximately equal amounts, though requiring increasingly smaller yaw angles, and by the wake of the last turbine, which deflects considerably even though the turbine is at almost zero yaw offset.

The inclination towards large yaw angles at the first few turbines may be attributed to the actuator disk theory overestimating the power of yawed turbines. Nevertheless, the operational details of the turbines are identical in both flow models (LES-ADM and FLORIS). This suggests that, in the comparison between the two flow models, the preference for larger yaw angles in the LES is likely related to the flow mechanisms (e.g., row interactions) that are missing in FLORIS. The above have now been added as comments in section 4.2 in the revised manuscript.

Numerical blockage:
For the second study, the authors assess that the estimated efficiency would reduce from 4% to 3.3% by reducing the numerical blockage from $(\pi \cdot 80m^2)/(560m \cdot 1024m) = 3.5\%$ to $(\pi \cdot 80m^2)/(5 \cdot 560m \cdot 1024m) = 0.7\%$ (I assume it is a typo that line 282 states km, not m). This analysis is good and the results plausible. However, in the first study, the numerical blockage is up to $(16 \cdot \pi \cdot 100m^2)/(3340m \cdot 501m) = 30\%$ and even if the 16 turbines were arranged in a $4 \times 4$ it would correspond to 7.5%. For such layouts, the numerical blockage is significant and the impact clearly seen in the results with substantial speedups, where certain turbines have efficiencies of 110% (Figure 7). I am sceptical how realistic these speed-ups are, particular given the sensitivity mentioned for the second study. The article states that the impact is less than 1% when tested with a domain three times wider (line 230-231) and refers to Antonini et al. (2018) and King et al. (2017). I can not find the details in Antonini et al., which is also a 2D simulation, but King et al. have blockage ratio of 5.2% for worst case scenario and reference Chen and Liou to give a threshold of less than 10% for wind tunnel studies. I think the details of this analysis is required in an appendix.

In both cases care was taken so that blockage effects are limited (to a degree that would still render performing all simulations feasible). In section 3, the maximum blockage (if all turbines are aligned

normally to the flow, and without considering the porosity of the turbines) is $(16 \cdot \pi R^2)/(L_y L_z) = (16 \cdot \pi \cdot 50^2)/(3340 \cdot 501) = 7.5\%$. In the second study, the blockage is $(\pi \cdot 40^2)/(560 \cdot 1024) \leq 1\%$. (The values provided by the reviewer were computed using the diameter rather than the radius of the turbines.) The domain blockage in all simulations remains below the 10% threshold mentioned by the reviewer. The typo (km instead of m) has been corrected.

Uncertainties:
- line 126: In principle, turbines should align themselves with the incoming wind direction. The reality is however, that unintended yaw misalignment is a very large uncertainty in normal operating wind farms, and it is a large challenge in order to actually apply wind farm control, where it is notoriously difficult to determine a wind direction and hence provide accurate estimates of how much to yaw, see for instance [5, 6].

The following comment has been added in the revised manuscript to reflect this issue:
"The turbines can align themselves with the mean direction of incoming wind. In practice, accurate and robust estimation of the wind direction poses a significant challenge [1]."

- Section 2.1 and line 206: SLSQP is used to optimize FLORIS while LES is optimized with BO, but does the choice of optimization strategy not potentially affect the optimization results? Please motivate why different methods are used.

SLSQP is used as it is the default and most well-tested optimisation method in FLORIS (this is now stated in section 3.2 and Appendix B of the revised manuscript). [11] studied the performance of different optimisation methods and showed that they perform similarly when appropriately used. One potential issue with gradient-based methods as SLSQP is their poor performance when dealing with multi-modal optimisation problems. To address this in the present study and enable a comprehensive comparison with BO (which is a global optimisation method), we implemented a multi-start approach for layout optimisation, and utilised a set of optimal yaw angles (as obtained in [6]) as initial conditions for the wake steering problem.
On the other hand, the optimisation framework we propose relies on LES data. As discussed in sections 1 and 2 in the manuscript, a Bayesian approach is adopted as it is particularly attractive for optimising expensive-to-calculate functions and for use in problems where adjoint methods would be hard to apply.

- There are significant uncertainties of wake steering related to the impact on structural loads, particular for large yaw angles, which is not addressed.

A relevant comment along with two references has been added in section 4.2 of the revised manuscript:
"For large yaw offsets, it is important to also consider the effects of yawing the turbine on the loads it experiences [4, 5]."

Unclear comments:
- Line 21: I think it is misleading to say that wind farms become increasingly less efficient as their size increase. First, many studies have shown how there an equilibrium between power extracted

by turbines and the entrained energy from the atmosphere, see e.g. [7]. Second, wind farm layouts have historically developed from aligned/rectangular layouts to curved and finally more misaligned and "random" layouts. The authors use Horns Rev for the second study, and this development in layouts is very clear in the three different generations of wind farms at Horns Rev, see figure 1. The figure is reproduced from [8], where it can also be seen (Table 2) how the capacity factor has increased historically for these wind farms.

The sentence has now been changed to:
"One key issue is that wind farms that consist of many turbines are typically less efficient than smaller wind farms."

- line 34: Wake steering is still not generally applied on commercial wind farms, despite growing scientific evidence. The main reason is that the uncertainties related to wake steering remains very high, i.e. it is not given that overall power output will increase, see for instance the review paper [9] or the benchmark paper [10]. I think the article would benefit for including such considerations at least in the motivation and discussion.

The relevant discussion in the introduction has now been changed to:
"Demonstrations of performance gains in a number of computational (Fleming et al., 2015; Gebraad et al., 2016), experimental (Adaramola and Krogstad, 2011; Campagnolo et al., 2016; Bastankhah and Porté-Agel, 2019) and field (Fleming et al., 2017; Howland et al., 2019; Fleming et al., 2019; Simley et al., 2021) studies have prompted the consideration of implementing wake steering in commercial wind plants (see, for instance, a press release by WindESCo (2023)). Nevertheless, further research is required in order to reduce the uncertainties surrounding its potential benefits (Kheirabadi and Nagamune, 2019)."

- line 55: RANS is fully capable of modeling wake-to-wake interaction.

The revised sentence now reads:
"However, the unsteady wake dynamics and the atmosphere-to-wake interactions, which play a critical role in large wind farm flows, cannot be appropriately accounted for."

- Figure 6 and text: I find the text unclear related to the figure. First, in line 215 it says that 70% of the designs found by LES-BO produce more power than the designs found by FLORIS. Looking at Figure 6, LES-BO (black line) is alway above FLORIS (red line), i.e. 100%. The 70% seems to be that the combination of LF/LES outperforms LES-BO. Line 219 is also unclear as to which wake model outperforms LES-BO, but it appears to be LF/LES. It is not clear in the figure, but have FLORIS and LF/LES been sorted independently or does design "n" of FLORIS correspond to design "n" of LF/LES? If not, it would be interesting to see a correlation plot of the two.

The sentence in question has been rephrased to make this point clear to the reader:
"First, the proposed LES-BO framework is capable of finding a design that produces more power than $\approx 70\%$ of the optimal designs suggested by `FLORIS` (as evaluated with LES)".
The layouts have been sorted from best to worst based on the LF/LES data (this is now clarified in the revised manuscript). Figure 6 has also been updated, as we had erroneously used a sorted

variant of the matrix containing the FLORIS data in the production of the figure. As noted in the revised manuscript, it may now be observed that though qualitatively similar, there is no one-to-one correspondence between the predictions of the two flow models.

- Section 4: The two studies are somewhat disjointed, and could perhaps even have been clearer in two independent articles. Why not optimize for yaw angle on the optimal wind farm layout of Section 3?

The wake steering optimisation study was selected because the particular scenario showed very large sensitivity when optimised with wake models (see, for example, [13, 6]). Attempts at sequentially or jointly optimising farm layout and wake steering with the framework proposed in this work will be made in future work.

Minor Comments and Technical Corrections:
- Figure 5: Please define "f" (y-axis)
- Figure 9: Please add turbine numbers.
- Figure 10: What does the hatched bars indicate?

- $f$ is now defined in the caption of figure 5.
- Done.
- The hatched bars indicate the efficiency of the design suggested by the multifidelity approach (labelled EI-LES-BO, see also the legend in Figure 10).

**References**

[1] J. Annoni, C. Bay, K. Johnson, E. Dall'Anese, E. Quon, T. Kemper, and P. Fleming. Wind direction estimation using scada data with consensus-based optimization. *Wind Energy Science*, 4(2):355–368, 2019.

[2] N. Bempedelis, S. Laizet, and G. Deskos. Turbulent entrainment in finite-length wind farms. *Journal of Fluid Mechanics*, 955:A12, 2023.

[3] M. Calaf, C. Meneveau, and J. Meyers. Large eddy simulation study of fully developed wind-turbine array boundary layers. *Physics of Fluids*, 22(1), 2010.

[4] R. Damiani, S. Dana, J. Annoni, P. Fleming, J. Roadman, J. van Dam, and K. Dykes. Assessment of wind turbine component loads under yaw-offset conditions. *Wind Energy Science*, 3(1):173–189, 2018.

[5] B. L. Ennis, J. R. White, and J. A. Paquette. Wind turbine blade load characterization under yaw offset at the swift facility. In *Journal of Physics: Conference Series*, volume 1037, page 052001. IOP Publishing, 2018.

[6] F. Gori, S. Laizet, and A. Wynn. Sensitivity analysis of wake steering optimisation for wind farm power maximisation. *Wind Energy Science*, 8(9):1425–1451, 2023.

[7] K. S. Heck, H. M. Johlas, and M. F. Howland. Modelling the induction, thrust and power of a yaw-misaligned actuator disk. *Journal of Fluid Mechanics*, 959:A9, 2023.

[8] M. F. Howland, J. Bossuyt, L. A. Martínez-Tossas, J. Meyers, and C. Meneveau. Wake

structure in actuator disk models of wind turbines in yaw under uniform inflow conditions. *Journal of Renewable and Sustainable Energy*, 8(4), 2016.

[9] C. Jané-Ippel, N. Bempedelis, R. Palacios, and S. Laizet. Bayesian optimisation of a two-turbine layout around a 2D hill using Large Eddy Simulations. In preparation.

[10] G. A. Speakman, M. Abkar, L. A. Martínez-Tossas, and M. Bastankhah. Wake steering of multirotor wind turbines. *Wind Energy*, 24(11):1294–1314, 2021.

[11] J. J. Thomas, N. F. Baker, P. Malisani, E. Quaeghebeur, S. Sanchez Perez-Moreno, J. Jasa, C. Bay, F. Tilli, D. Bieniek, N. Robinson, A. P. J. Stanley, W. Holt, and A. Ning. A comparison of eight optimization methods applied to a wind farm layout optimization problem. *Wind Energy Science*, 8(5):865–891, 2023.

[12] Y.-T. Wu and F. Porté-Agel. Modeling turbine wakes and power losses within a wind farm using LES: An application to the Horns Rev offshore wind farm. *Renewable Energy*, 75:945–955, 2015.

[13] H. Zong and F. Porté-Agel. Experimental investigation and analytical modelling of active yaw control for wind farm power optimization. *Renewable Energy*, 170:1228–1244, 2021.

---

## Author Comment (AC2)

**Reply to reviewer #2**
**Manuscript wes-2023-110**

**Nikolaos Bempedelis, Filippo Gori, Andrew Wynn, Sylvain Laizet, and Luca Magri**

The remarks of the reviewer are repeated in blue point by point followed by our answers, while the main modifications in the paper are red-marked in the revised manuscript.

Summary:
The authors describe an optimization method that uses data from low- and high-fidelity models employing a Bayesian framework. The authors test their optimization method in maximizing wind farm power production through micro-siting and wake steering. The authors compare the best-performing wind farm layout design from their optimization model against optimized wind farm layouts using FLORIS. The proposed framework can generate layouts with similar wind farm efficiency when compared to the optimized layouts obtained using FLORIS. For wake steering, the proposed optimization framework can overperform as compared to optimization using FLORIS. The LES-informed framework can leverage the high-fidelity model capabilities in capturing complex flow features for wind turbine siting and wake steering. The manuscript is well written, and the results are very interesting. However, I recommend major revisions to incorporate important details in the methodology and results.

We thank the reviewer for the time devoted to reading our manuscript and for providing helpful comments and feedback.

Major comments:
Incomplete description of LES framework: The authors perform an impressive number of large-eddy simulations, but the description of the model setup is lacking. The authors are simulating atmospheric flow, but do they incorporate Coriolis in their simulations? Is there a capping inversion in their model, or is the potential temperature profile constant over the entire domain? What are the boundary conditions for the LES used for wind farm layout optimization (Monin-Obukhov similarity at the surface? periodic lateral BC?)? The actuator disk model uses a constant thrust coefficient (not realistic), but how is turbine power estimated (especially for partially waked conditions, like in Figure 2)? The turbine's thrust coefficient changes with yaw angle (Gebraad et al., 2017), which might partially explain the extreme yaw misalignment for the first three turbines in Section 4.

We thank the reviewer for their analysis and suggestions. The atmospheric boundary layer that is fed as inflow to the wind farm simulations is a fully-developed, neutral (no thermal effects), turbulent boundary layer that is driven by an imposed pressure gradient (no geostrophic wind forcing and Coriolis effects). [2] offers a very insightful discussion on the similarities and differences between the two forcing approaches.

The boundary conditions for the layout optimisation problem are: periodic in the lateral boundaries, free-slip at the top boundary, and no-slip with a wall-model at the ground.

The actuator disk model computes the power as $P = Tu_d$ where $T$ is the computed thrust and $u_d$ is the temporally-filtered disk-averaged velocity normal to the disk plane. In our implementation (which considers a constant "local" or "modified" thrust coefficient $C_T'$, see [2, 1]), both thrust and

power scale with the rotor-normal velocity, and are therefore functions of the turbine yaw angle. (The respective coefficients hence also change with the yaw angle, see also [5]). The predictions of our solver for the evolution of thrust and power with changing yaw angle [4] are matching those of other codes that employ the standard actuator disk method (see, for example, [5, 3]).

The above have been added as comments in an extended discussion of the LES framework in sections 2.2 and 3.1 of the revised manuscript.

Blockage and speedups: The authors report that front-row turbines produce less power than a stand-alone turbine due to blockage, and that downstream turbines can produce more power than a stand-alone turbine due to speedups. I think these statements need to be explained further. Bleeg and Montavon (2022) show the importance of including a capping inversion in the simulation domain and the sensitivity to domain size for simulating blockage. Regarding speedups, the maximum wind speed in Figure 4 appears to be close to 9 m s$^{-1}$, which is an $\approx 8\%$ speedup compared to freestream conditions. Furthermore, some downstream turbines are producing $\approx 10\%$ more power than a standalone turbine. These speedups can be an artifact of the width of the numerical domain. How did these speedup regions change when you tested the 3 times wider numerical domain?

The case where we tested the three times wider numerical domain is the one shown in Figure 7. In that case, the maximum streamwise velocity at hub height for the regular and extended domains was 8.845 and 8.807 m s$^{-1}$, respectively. This suggests that although limited, domain blockage effects are still present. In the revised manuscript, we have modified the discussion in section 3.2 to reflect the above:

"Here, an additional simulation with three times larger spanwise extent was performed to evaluate the effects of domain blockage. These were confirmed to be present but relatively small, with a $\approx 0.43\%$ difference in maximum streamwise velocity). Similar benefits owing to local blockage were also reported by King et al. (2017) and Antonini et al. (2018). Nevertheless, it is important to highlight that blockage effects are particularly sensitive to the atmospheric conditions besides the extent of the computational domain (Bleeg and Montavon, 2022)."

Computational requirements of this approach: The authors compare the optimized layouts obtained from LES- and FLORIS-informed frameworks, showing that the LES can produce better results about 70% of the time. It is important to highlight the computational requirements needed to perform the LES- and FLORIS-informed optimizations given that the layouts from FLORIS can overperform when compared to the LES-BO methodology. Furthermore, how realistic is performing 4200 LES for wind turbine siting as compared to optimizing the layout using FLORIS and then evaluating multiple possible layouts using LES?

To highlight the effectiveness of wake models in the layout optimisation problem (further to the discussion in section 3.2 of the manuscript), we have added the following comment in section 5 (Conclusions):

"...The performance of wake models was also found to be outstanding. This is particularly important, given the excessive computational requirements of LES-based layout optimisation."

The large computational demands of LES-based optimisation prevent it from being directly applicable in engineering design practices. However, the potential benefits in scenarios of increased flow complexity point towards the development of multi-fidelity methods as a particularly promising

line of research.

The authors show the capability of their methodology for optimizing a wind farm's layout and wake steering for a single turbine row. Can these two problems be addressed in the same optimization problem? Also, how feasible is it to optimize the yaw angles for wake steering for a whole wind farm rather than for a single turbine row?

The two problems can be addressed simultaneously by the proposed method. However, increasing the number of design variables (to three design variables per turbine, or potentially even more) is associated with an increase in the number of evaluations required, and thus further increases in the computational cost. Therefore, in large optimisation problems, multi-fidelity methods may be necessary to mitigate parts of the cost.
Optimising the yaw angles for the entire Horns Rev I wind farm would require a considerably larger number of evaluations, with each simulation also being costlier (a larger spanwise domain would be required to accommodate the additional rows with minimal blockage). While we believe this task could be realisable, especially given we can begin the entire farm optimisation from a well-informed state, we expect that the benefits may not justify the additional costs.

Minor comments:
Figure 4: Rotating the reference frame in Figure 4 can be confusing for the reader. It might seem as if multiple layouts are being tested rather than a single layout for multiple wind directions.

The caption of figure 4 has been modified to make this clear to the reader:
"The flow fields show the same layout exposed to different wind directions. In all cases, the wind is shown as blowing from left to right, ..."

What are the intermittent vertical lines in Figure 9 that appear in front of some turbines (e.g., turbines 11, 12, 13, 14, 16)?

These are numerically-introduced artifacts owing to the sharp and discontinuous nature of the passive scalar sources.

FLORIS can incorporate varying thrust coefficients for waked turbines. Did you try incorporating a thrust curve in your actuator disk model so that the velocity deficit in waked turbines is not underestimated?

Both FLORIS and the LES-ADM framework may incorporate information from thrust and power coefficient curves (see, for example, [6]). In the present study, we opted for the simplifiyng assumption of constant coefficients in both frameworks. However, this assumption could be addressed in future work.

**References**

[1] N. Bempedelis, S. Laizet, and G. Deskos. Turbulent entrainment in finite-length wind farms. *Journal of Fluid Mechanics*, 955:A12, 2023.

[2] M. Calaf, C. Meneveau, and J. Meyers. Large eddy simulation study of fully developed wind-turbine array boundary layers. *Physics of Fluids*, 22(1), 2010.

[3] K. S. Heck, H. M. Johlas, and M. F. Howland. Modelling the induction, thrust and power of a yaw-misaligned actuator disk. *Journal of Fluid Mechanics*, 959:A9, 2023.

[4] C. Jané-Ippel, N. Bempedelis, R. Palacios, and S. Laizet. Bayesian optimisation of a two-turbine layout around a 2D hill using Large Eddy Simulations. In preparation.

[5] G. A. Speakman, M. Abkar, L. A. Martínez-Tossas, and M. Bastankhah. Wake steering of multirotor wind turbines. *Wind Energy*, 24(11):1294–1314, 2021.

[6] Y.-T. Wu and F. Porté-Agel. Modeling turbine wakes and power losses within a wind farm using les: An application to the horns rev offshore wind farm. *Renewable Energy*, 75:945–955, 2015.

---

## Referee Report (RR1)

**2nd Review of:**
**"Data-driven optimisation of wind farm layout and wake steering with large-eddy simulations"**
**by Nikolaos Bempedelis, Filippo Gori, Andrew Wynn, Sylvain Laizet, and Luca Magri**

New comments have been added in red.

**General Comments:**

The article present two different studies on optimization of wind farm performance, one on layout optimization and one on wind farm control using wake steering. The optimizations are performed using high-fidelity (LES) and low fidelity (FLORIS) wake modelling as well as multi-fidelity by combining results from both. The results are based on an impressive amount of LES, and provide interesting results. However, the article also show several shortcomings and several aspects, which lack important details for instance in terms of modeling choices, numerical blockage and uncertainties. Therefore, I recommend major revisions.

We thank the reviewer for the time devoted to reading our manuscript and for providing helpful comments and feedback.

The provided revision has minor adjustments to previous comments, but does no properly address a number of my comments from the first review and some new comments/results appear misleading. This is at least partly due to important details still missing from the manuscript, which still leaves a reader guessing or assuming key details of what the authors have done.

**Specific Comments:**

**Lack of details:**

**Actuator Disc modeling and Power estimation:**

The description of the actuator disc method is very inadequate, and Bempedelis et al. 2023 does not provide the necessary details. I can only assume that an uniformly loaded actuator disc is employed. If so, it is far from state-of-the-art and it has been shown several times to be insufficient, particular for more complex flows such as yawing turbines, see for instance [1], where a standard (uniform) actuator disc is compared to a BEM based actuator disc with rotation.

[1] also show how power production behind a uniformly loaded actuator disc is significantly over-estimated compared to a slightly more advanced actuator disc model using BEM and rotation.

Therefore, the authors should specify how the power is estimated, as this is particular important for yawing turbines, and further complicated if yawing turbines are operating in waked conditions, see [2]. I suspect the efficiencies in Figure 12 are a result of simplification, but it is hard to test/decipher as a reader. It would be beneficial to also report the actual power production (in [kW]), not only the efficiency. Furthermore, I have some concerns about the optimization angles shown in Figure 12. First, it is a general concern when optimizations consistently provides results at the admissible limit, as is the case where the first four turbines are yawing 30°. Second, several higher fidelity simulations have shown that there are secondary effects of wake steering. This means the turbines operating in wake will typically yaw less, see [3] for two turbines, where the second turbine yaws positively to increase production, and [4], where deep-row turbines should yaw relatively less. Such secondary effects are actually also visible in Figure 8.

We thank the reviewer for the thorough analysis and comment. The wind turbines are modelled with a standard non-rotational actuator disk model. In section 2.2 of the revised manuscript, we have added a brief description of how power is computed in the actuator disk method, along with four more references [3, 10, 7, 9] besides [2]. Additionally, we have added a link that directs the interested reader to the open-source code repository for direct access to all methodology and implementation details (https://github.com/xcompact3d).

As suggested by the reviewer, the ADM with rotation is a more advanced turbine representation model compared to the standard ADM that we employ in the present study. Nevertheless, the optimisation framework we propose is compatible with any type of turbine representation model (e.g., standard actuator disk, actuator disk with rotation, or actuator line). This is now mentioned in section 2.2 of the revised manuscript, together with a comment on the potential for improved turbine wake and power predictions through use of more advanced turbine modelling approaches. The efficiencies shown in figure 12 are calculated as $\eta_{ind,i} = P_i(\gamma)/P_i(\gamma = 0°)$ (see section 4.2 in the manuscript). Efficiencies are used as a more suitable metric for the comparisons made in this study, as opposed to presenting dimensional power values. The actuator disk method inherently presumes a connection between thrust and power coefficients [10] unless the user specifies otherwise [12]. In our study, where constant thrust (and power) coefficients are assumed, employing efficiency metrics effectively conveys the trends without the need to specify a power coefficient that is representative of a specific turbine only.

Secondary steering effects are accounted for in our simulations. LES-ADM naturally captures the counter-rotating vortices that are present in the wake of a yawed turbine and lead to its curled shape (see, for example, [8]). This is evidenced by figures 12 and 13, where the wakes of downstream turbines deflect by approximately equal amounts, though requiring increasingly smaller yaw angles, and by the wake of the last turbine, which deflects considerably even though the turbine is at almost zero yaw offset. The inclination towards large yaw angles at the first few turbines may be attributed to the actuator disk theory overestimating the power of yawed turbines. Nevertheless, the operational details of the turbines are identical in both flow models (LES-ADM and FLORIS). This suggests that, in the comparison between the two flow models, the preference for larger yaw angles in the LES is likely related to the flow mechanisms (e.g., row interactions) that are missing in FLORIS. The above have now been added as comments in section 4.2 in the revised manuscript.

In the reply to my review and the other reviewer, the implementation of the actuator disc is still lacking details. The authors state that CT is constant (e.g. line 170), but it is unclear how and if CT changes when yawing. In the reply to the other reviewer the authors repeat that CT is constant, but also state that it is function of yaw angle. This is quite critical for the present study. If CT is indeed constant, is the total thrust force also constant? If so, then a yawing turbine would reduce the streamwise induction (as the thrust is directed with an angle laterally), leading to an small increase in the mean streamwise wind speed and therefore increase in power P = u*T with constant T. That would mean that yawing turbines are not penalized in terms of power production.

The LES results would essentially be meaningless and should not be published. I do not think this is what is going on, particular as the authors state that the efficiency of the turbines depends on the yaw angle, but as it is, I have to speculate and assume this. The authors need to describe this in proper detail for the readers, for example using an exponent to describe changes in CT and CP as in [2], which as mentioned in my first review is particular important for yawing turbines in wake. I'm not convinced that "efficencies are ...more suitable metric" as [1] clearly shows that the efficiency is overestimated with this actuator disc implementation. Reporting efficiencies for the simplified actuator disc makes it harder for readers to actually decipher the results. Again, having to guess/speculate on the underlying numbers due to lack of details. However, with proper explanation of how the turbines are modelled in yaw it might be OK.

Finally, the authors do not address my concern about the yaw angles of the first four turbines in LES-BO being optimized to the limit of the allowable range ($30°$). This is clearly shows that the optimization limits are not wide enough, which is a basic flaw in optimization. The authors should address this.

**Numerical blockage:**

For the second study, the authors assess that the estimated efficiency would reduce from 4% to 3.3% by reducing the numerical blockage from $(\pi \cdot 80m^2)/(560m \cdot 1024m) = 3.5\%$ to $(\pi \cdot 80m^2)/(5 \cdot 560m \cdot 1024m) = 0.7\%$ (I assume it is a typo that line 282 states km, not m). This analysis is good and the results plausible. However, in the first study, the numerical blockage is up to $(16 \cdot \pi \cdot 100m^2)/(3340m \cdot 501m) = 30\%$ and even if the 16 turbines were arranged in a $4 \times 4$ it would correspond to 7.5%. For such layouts, the numerical blockage is significant and the impact clearly seen in the results with substantial speedups, where certain turbines have efficiencies of 110% (Figure 7). I am sceptical how realistic these speed-ups are, particular given the sensitivity mentioned for the second study. The article states that the impact is less than 1% when tested with a domain three times wider (line 230-231) and refers to Antonini et al. (2018) and King et al. (2017). I can not find the details in Antonini et al., which is also a 2D simulation, but King et al. have blockage ratio of 5.2% for worst case scenario and reference Chen and Liou to give a threshold of less than 10% for wind tunnel studies. I think the details of this analysis is required in an appendix.

In both cases care was taken so that blockage effects are limited (to a degree that would still render performing all simulations feasible). In section 3, the maximum blockage (if all turbines are aligned normally to the flow, and without considering the porosity of the turbines) is $(16 \cdot \pi R2)/(LyLz) = (16 \cdot \pi \cdot 502)/(3340 \cdot 501) = 7.5\%$. In the second study, the blockage is $(\pi \cdot 402)/(560 \cdot 1024) \le 1\%$. (The values provided by the reviewer were computed using the diameter rather than the radius of the turbines.) The domain blockage in all simulations remains below the 10% threshold mentioned by the reviewer. The typo (km instead of m) has been corrected.

I apologize that I made an error in my previous assessment of the blockage. However, I still suspect that the reply and additional text by the authors is misleading. Evaluating blockage in terms of the difference in the ***maximum streamwise velocity*** within the domain is not representative of the impact of numerical blockage on the operation and power production of the wind farm. I suspect/guess the small increase in maximum velocity occurs high in the domain, where the velocity is already high due to the atmospheric boundary layer. In order to assess the numerical blockage, you should report speed ups of mean wind speed at the turbines and differences in power production with the two different domains, one 3 times wider than the other. Typically, physical wind farm blockage is results in 1-2% changes in power, see e.g. [5, 6], not 5-8% as shown in Figure 7.

**Uncertainties:**

- line 126: In principle, turbines should align themselves with the incoming wind direction. The reality is however, that unintended yaw misalignment is a very large uncertainty in normal operating wind farms, and it is a large challenge in order to actually apply wind farm control, where it is notoriously difficult to determine a wind direction and hence provide accurate estimates of how much to yaw, see for instance [7, 8].

The following comment has been added in the revised manuscript to reflect this issue: "The turbines can align themselves with the mean direction of incoming wind. In practice, accurate and robust estimation of the wind direction poses a significant challenge [1]."

I find it a bit hard to understand this sentence and where it is placed in the text. Does "The turbines can align themselves..." refer to the optimisation setup or to reality? If it relates to the simulations and the setup, then it implies that the authors have utilized a numerical wind direction controller. If so, is the reference baseline before optimizing the yaw angles including alignment to local wind directions for all turbines? The footnote clearly relates to reality, but placed in the setup of the optimisation. I suggest to rephrase to correctly represent if the simulations does or does not include a wind direction controller utilized to determine the baseline, and that this is a challenge in reality. I suggest to place such comments in a more appropriate section, for instance while discussing limitations and uncertainties of the present study.

- Section 2.1 and line 206: SLSQP is used to optimize FLORIS while LES is optimized with BO, but does the choice of optimization strategy not potentially affect the optimization results? Please motivate why different methods are used.

SLSQP is used as it is the default and most well-tested optimisation method in FLORIS (this is now stated in section 3.2 and Appendix B of the revised manuscript). [11] studied the performance of different optimisation methods and showed that they perform similarly when appropriately used. One potential issue with gradient-based methods as SLSQP is their poor performance when dealing with multi-modal optimisation problems. To address this in the present study and enable a comprehensive comparison with BO (which is a global optimisation method), we implemented a multi-start approach for layout optimisation, and utilised a set of optimal yaw angles (as obtained in [6]) as initial conditions for the wake steering problem. On the other hand, the optimisation framework we propose relies on LES data. As discussed in sections 1 and 2 in the manuscript, a Bayesian approach is adopted as it is particularly attractive for optimising expensive-to-calculate functions and for use in problems where adjoint methods would be hard to apply.

OK.

- There are significant uncertainties of wake steering related to the impact on structural loads, particular for large yaw angles, which is not addressed.

A relevant comment along with two references has been added in section 4.2 of the revised manuscript: "For large yaw offsets, it is important to also consider the effects of yawing the turbine on the loads it experiences [4, 5]."

OK. The authors have added a few scattered comments on uncertainty, but the uncertainties remain in the presented results, and comments on uncertainties and limitations remains limited. Therefore, I suggest the authors to be less certain and take more reservations to their own results, for instance on wake models performing "outstanding" compared to an simplified turbine representation in CFD. In my opinion, it is good scientific practice to be critical and outline limitations and uncertainties of ones results. It does not take away value, but builds confidence and facilitates scientific discussion.

**Unclear comments:**

- Line 21: I think it is misleading to say that wind farms become increasingly less efficient as their size increase. First, many studies have shown how there an equilibrium between power extracted by turbines and the entrained energy from the atmosphere, see e.g. [9]. Second, wind farm layouts have historically developed from aligned/rectangular layouts to curved and finally more misaligned and "random" layouts. The authors use Horns Rev for the second study, and this development in layouts is very clear in the three different generations of wind farms at Horns Rev, see figure 1. The figure is reproduced from [10], where it can also be seen (Table 2) how the capacity factor has increased historically for these wind farms.

The sentence has now been changed to: "One key issue is that wind farms that consist of many turbines are typically less efficient than smaller wind farms."

The sentence has been rephrased, but the meaning is the same: Larger wind farms are less efficient than small. This is still misleading and I think this statement requires a solid reference, as I have provided reference which shows the opposite and as commented in my previous review this is (to my knowledge) not general knowledge and stands as an unsupported statement.

- line 34: Wake steering is still not generally applied on commercial wind farms, despite growing scientific evidence. The main reason is that the uncertainties related to wake steering remains very high, i.e. it is not given that overall power output will increase, see for instance the review paper [11] or the benchmark paper [12]. I think the article would benefit for including such considerations at least in the motivation and discussion.

The relevant discussion in the introduction has now been changed to: "Demonstrations of performance gains in a number of computational (Fleming et al., 2015; Gebraad et al., 2016), experimental (Adaramola and Krogstad, 2011; Campagnolo et al., 2016; Bastankhah and Porté-Agel, 2019) and field (Fleming et al., 2017; Howland et al., 2019; Fleming et al., 2019; Simley et al., 2021) studies have prompted the consideration of implementing wake steering in commercial wind plants (see, for instance, a press release by WindESCo (2023)). Nevertheless, further research is required in order to reduce the uncertainties surrounding its potential benefits (Kheirabadi and Nagamune, 2019)."

OK.

- line 55: RANS is fully capable of modeling wake-to-wake interaction.

The revised sentence now reads: "However, the unsteady wake dynamics and the atmosphere-to-wake interactions, which play a critical role in large wind farm flows, cannot be appropriately accounted for."

OK.

- Figure 6 and text: I find the text unclear related to the figure. First, in line 215 it says that 70% of the designs found by LES-BO produce more power than the designs found by FLORIS. Looking at Figure 6, LES-BO (black line) is alway above FLORIS (red line), i.e. 100%. The 70% seems to be that the combination of LF/LES outperforms LES-BO. Line 219 is also unclear as to which wake model outperforms LES-BO, but it appears to be LF/LES. It is not clear in the figure, but have FLORIS and LF/LES been sorted independently or does design "n" of FLORIS correspond

to design "n" of LF/LES? If not, it would be interesting to see a correlation plot of the two.

The sentence in question has been rephrased to make this point clear to the reader: "First, the proposed LES-BO framework is capable of finding a design that produces more power than $\approx 70\%$ of the optimal designs suggested by FLORIS (as evaluated with LES)". The layouts have been sorted from best to worst based on the LF/LES data (this is now clarified in the revised manuscript). Figure 6 has also been updated, as we had erroneously used a sorted variant of the matrix containing the FLORIS data in the production of the figure. As noted in the revised manuscript, it may now be observed that though qualitatively similar, there is no one-to-one correspondence between the predictions of the two flow models.

[Figure]

Thanks for updating the sorting. I think it provides very interesting insights as to how well the simple wake model actually performs. .

- Section 4: The two studies are somewhat disjointed, and could perhaps even have been clearer in two independent articles. Why not optimize for yaw angle on the optimal wind farm layout of Section 3?

Figure 1: Layout of three generations of wind farms at Horns Rev.

The wake steering optimisation study was selected because the particular scenario showed very large sensitivity when optimised with wake models (see, for example, [13, 6]). Attempts at sequentially or jointly optimising farm layout and wake steering with the framework proposed in this work will be made in future work.

OK.

**Minor Comments and Technical Corrections:**

- Figure 5: Please define "f" (y-axis)
 f is now defined in the caption of figure 5.

"f" is now defined as "the standardised overall farm power output", but this is not a standard definition nor is it clear. Please provide a definition in terms of an equation how "f" is computed.

- Figure 9: Please add turbine numbers.
  Done.

OK.

- Figure 10: What does the hatched bars indicate?

The hatched bars indicate the efficiency of the design suggested by the multifidelity approach (labelled EI-LES-BO, see also the legend in Figure 10).

My point was that only two of the six EI-LES-BO bars are hatched. What is the difference between hatched and non-hatched bars? If no difference, please provide consistent plotting.

Additional comments:
- Line 179: Boundary conditions of the simulation setup is detailed in both Sections 2.2 and 3.1. I suggest to move the LES description to Section 2.2.
- Figure 2: I'm not familiar with how precisely regulators define areas and if it is based on location of towers or full extend of rotor, but it is clearly visible in Figure 2 that the rotor of the top turbine extends outside of the available area. Please comment.
- Figure 7: Please extend the y-axis to cover the full range so readers can assess increase in efficiencys above 100.

**References**

[1] Lin M and Porté-Agel F 2019 *Energies* **12** ISSN 1996-1073

[2] Liew J, Urbán A M and Andersen S J 2020 *Wind Energy Science* **5** 427–437 URL `https://wes.copernicus.org/articles/5/427/2020/`

[3] Hulsman P, Andersen S J and Göçmen T 2020 *Wind Energy Science* **5** 309–329

[4] Archer C L and Vasel-Be-Hagh A 2019 *Sustainable Energy Technologies and Assessments* **33** 34–43 ISSN 2213-1388

[5] Forsting A R M and Troldborg N 2015 *Journal of Physics: Conference Series* **625** 012029

[6] Meyer Forsting A R, Troldborg N and Gaunaa M 2017 *Wind Energy* **20** 63–77

[7] Quick J, Annoni J, King R, Dykes K, Fleming P and Ning A 2017 *Journal of Physics: Conference Series* **854** 012036 URL `https://dx.doi.org/10.1088/1742-6596/854/1/012036`

[8] Annoni J, Bay C, Johnson K, Dall'Anese E, Quon E, Kemper T and Fleming P 2019 *Wind Energy Science* **4** 355–368 URL `https://wes.copernicus.org/articles/4/355/2019/`

[9] Calaf M, Meneveau C and Meyers J 2010 *Phys. Fluids* **22**

[10] Sørensen J N and Larsen G C 2021 *Energies* **14** ISSN 1996-1073 URL `https://www.mdpi.com/1996-1073/14/2/448`

[11] Kheirabadi A C and Nagamune R 2019 *J. Wind Engin. Ind. Aero.* **192** 45–73

[12] Göçmen T, Campagnolo F, Duc T, Eguinoa I, Andersen S J, Petrović V, Imširović L, Braunbehrens R, Liew J, Baungaard M, van der Laan M P, Qian G, Aparicio-Sanchez M, González-Lope R, Dighe V V, Becker M, van den Broek M J, van Wingerden J W, Stock A, Cole M, Ruisi R, Bossanyi E, Requate N, Strnad S, Schmidt J, Vollmer L, Sood I and Meyers J 2022 *Wind Energy Science* **7** 1791–1825 URL `https://wes.copernicus.org/articles/7/1791/2022/`

---

## Author Response (AR2)

**Reply to reviewer #1**
**Manuscript wes-2023-110**

**Nikolaos Bempedelis, Filippo Gori, Andrew Wynn, Sylvain Laizet, and Luca Magri**

The new or remaining remarks of the reviewer are repeated in blue point by point followed by our answers, while the main modifications in the paper are red-marked in the revised manuscript.

General Comments:
The provided revision has minor adjustments to previous comments, but does no properly address a number of my comments from the first review and some new comments/results appear misleading. This is at least partly due to important details still missing from the manuscript, which still leaves a reader guessing or assuming key details of what the authors have done.

We thank the reviewer for the time devoted to reading our manuscript and for providing helpful comments and feedback.

Specific comments:
Actuator Disc modeling and Power estimation:
In the reply to my review and the other reviewer, the implementation of the actuator disc is still lacking details. The authors state that CT is constant (e.g. line 170), but it is unclear how and if CT changes when yawing. In the reply to the other reviewer the authors repeat that CT is constant, but also state that it is function of yaw angle. This is quite critical for the present study. If CT is indeed constant, is the total thrust force also constant? If so, then a yawing turbine would reduce the streamwise induction (as the thrust is directed with an angle laterally), leading to an small increase in the mean streamwise wind speed and therefore increase in power P = u*T with constant T. That would mean that yawing turbines are not penalized in terms of power production. The LES results would essentially be meaningless and should not be published. I do not think this is what is going on, particular as the authors state that the efficiency of the turbines depends on the yaw angle, but as it is, I have to speculate and assume this. The authors need to describe this in proper detail for the readers, for example using an exponent to describe changes in CT and CP as in [2], which as mentioned in my first review is particular important for yawing turbines in wake. I'm not convinced that "efficencies are ...more suitable metric" as [1] clearly shows that the efficiency is overestimated with this actuator disc implementation. Reporting efficiencies for the simplified actuator disc makes it harder for readers to actually decipher the results. Again, having to guess/speculate on the underlying numbers due to lack of details. However, with proper explanation of how the turbines are modelled in yaw it might be OK. Finally, the authors do not address my concern about the yaw angles of the first four turbines in LES-BO being optimized to the limit of the allowable range (30°). This is clearly shows that the optimization limits are not wide enough, which is a basic flaw in optimization. The authors should address this.

We apologise for any lack of clarity in the original manuscript. The thrust predictions rely on a constant local (or "modified") thrust coefficient $C_T'$ and the local velocity normal to the disk $u_d$, $T = \frac{1}{2}\rho A C_T' u_d^2$. Further to section 2.2 where these quantities are introduced, we have changed the discussion throughout the manuscript to highlight the consideration of the constant *local* coefficient $C_T'$ (not the "nominal" one $C_T$) and the ability of our solver to predict power degradation as a

[Figure]

Figure 1: Power production of a yawed wind turbine with $C_T' = 4/3$, normalised by the power production of a yaw-aligned wind turbine.

function of rotor misalignment. In its most explicit and simple form (see e.g., [8]) the thrust can be expressed as:

$$T = \frac{1}{2}\rho A C_T' u_d^2 = \frac{1}{2}\rho A C_T(\gamma) U_\infty^2 \cos^2 \gamma$$

where an approximation for $C_T(\gamma)$ can be found in [7]. Figure 1 shows the solver's predictions for turbine power as a function of the yaw angle (data are from [6] and were kindly provided by Mr. Jané-Ippel), together with the LES data of [5] and the approximation used by `FLORIS`.

Finally, a comment has been added in section 4.2:
"We note that the yaw angles of the first four turbines in the LES-BO design are at the arbitrarily selected yaw limits defined in the problem set-up (see Sect. 4.1). This means that there is potential for further improvements should these limits be extended or removed. In practice however, for large yaw offsets, it is important to also consider the effects of yawing the turbine on the loads it experiences [3, 4].

I apologize that I made an error in my previous assessment of the blockage. However, I still suspect that the reply and additional text by the authors is misleading. Evaluating blockage in terms of the difference in the maximum streamwise velocity within the domain is not representative of the impact of numerical blockage on the operation and power production of the wind farm. I suspect/guess the small increase in maximum velocity occurs high in the domain, where the velocity is already high due to the atmospheric boundary layer. In order to assess the numerical blockage, you should report speed ups of mean wind speed at the turbines and differences in power production with the two different domains, one 3 times wider than the other. Typically, physical wind farm blockage is results in 1-2% changes in power, see e.g. [5, 6], not 5-8% as shown in Figure 7.

The maximum streamwise velocity was reported as it was relevant to a comment of the second reviewer. In the revised manuscript, we have clarified that it is measured at the hub height, which is a characteristic level for the wind turbines. Additionally, we report the difference in farm power

production from the simulations with the two different domains. This was found to be ≈ 1.36%. The large increases relative to the production of a single wind turbine are a consequence of local blockage (i.e., flow being locally accelerated due to streamtube contraction), similar to what might be observed in the second row of staggered wind farms (see e.g., [9]).

I find it a bit hard to understand this sentence and where it is placed in the text. Does "The turbines can align themselves..." refer to the optimisation setup or to reality? If it relates to the simulations and the setup, then it implies that the authors have utilized a numerical wind direction controller. If so, is the reference baseline before optimizing the yaw angles including alignment to local wind directions for all turbines? The footnote clearly relates to reality, but placed in the setup of the optimisation. I suggest to rephrase to correctly represent if the simulations does or does not include a wind direction controller utilized to determine the baseline, and that this is a challenge in reality. I suggest to place such comments in a more appropriate section, for instance while discussing limitations and uncertainties of the present study.

This refers to the fact that we have a six-directional wind rose (see, for example, figure 2 in the manuscript). The wind rose shows the mean direction of the incoming wind, and the sentence in question notes that the turbines have a zero yaw with respect to that *mean* direction (no real-time or local mean controllers are used). The sentence has now been changed to reflect the above: "The turbines are aligned with the mean inflow wind direction in all cases considered in this section (no controllers are considered). In reality, accurate and robust estimation of the wind direction poses a significant challenge [1]."

OK. The authors have added a few scattered comments on uncertainty, but the uncertainties remain in the presented results, and comments on uncertainties and limitations remains limited. Therefore, I suggest the authors to be less certain and take more reservations to their own results, for instance on wake models performing "outstanding" compared to an simplified turbine representation in CFD. In my opinion, it is good scientific practice to be critical and outline limitations and uncertainties of ones results. It does not take away value, but builds confidence and facilitates scientific discussion.

The sentence has been changed to "The performance of wake models was also found to be remarkable."

The sentence has been rephrased, but the meaning is the same: Larger wind farms are less efficient than small. This is still misleading and I think this statement requires a solid reference, as I have provided reference which shows the opposite and as commented in my previous review this is (to my knowledge) not general knowledge and stands as an unsupported statement.

Our statement refers to long (or deep) wind farms, i.e., that the twentieth row of a wind farm will typically produce less power than the fourth (row numbers are indicative). Then, making the farm larger by adding rows at its "end" would decrease its overall efficiency. The sentence in question has been changed to reflect the above: "One key issue is that wind farms that consist of many turbine rows are typically less efficient than less deep wind farms (i.e., downstream rows produce less power than upstream ones) [2]."

"f" is now defined as "the standardised overall farm power output", but this is not a standard definition nor is it clear. Please provide a definition in terms of an equation how "f" is computed.

The following has been added in the caption of figure 5:
"$f$ is the standardised overall farm power output, $f = \left(P_F - \mathrm{mean}\left(P_F\right)\right)/\sigma(P_F)$, with $P_F$ denoting the farm output."

My point was that only two of the six EI-LES-BO bars are hatched. What is the difference between hatched and non-hatched bars? If no difference, please provide consistent plotting.

All six EI-LES-BO bars are hatched. This might be an issue with the software used by the reviewer to read the document.

Additional comments:
- Line 179: Boundary conditions of the simulation setup is detailed in both Sections 2.2 and 3.1. I suggest to move the LES description to Section 2.2.
- Figure 2: I'm not familiar with how precisely regulators define areas and if it is based on location of towers or full extend of rotor, but it is clearly visible in Figure 2 that the rotor of the top turbine extends outside of the available area. Please comment.
- Figure 7: Please extend the y-axis to cover the full range so readers can assess increase in efficiencys above 100.

- The discussion in section 3.1 has been changed to: "The flow field planes stored from the precursor simulation (see Sect. 2.2) are used at the inlet, a convective condition is used at the outlet, and the conditions at the remaining boundaries are similar to the description in Sect. 2.2."
- The discussion in section 3.1 has been changed to: "The land where the turbine towers can be placed is a square of size $18D \times 18D$ (for this set-up, the blades can extend outside of the available area)".
- Done.

**References**

[1] J. Annoni, C. Bay, K. Johnson, E. Dall'Anese, E. Quon, T. Kemper, and P. Fleming. Wind direction estimation using scada data with consensus-based optimization. *Wind Energy Science*, 4(2):355–368, 2019.

[2] R. J. Barthelmie, K. Hansen, S. T. Frandsen, O. Rathmann, J. G. Schepers, W. Schlez, J. Phillips, K. Rados, A. Zervos, E. S. Politis, and P. K. Chaviaropoulos. Modelling and measuring flow and wind turbine wakes in large wind farms offshore. *Wind Energy*, 12(5):431–444, 2009.

[3] R. Damiani, S. Dana, J. Annoni, P. Fleming, J. Roadman, J. van Dam, and K. Dykes. Assessment of wind turbine component loads under yaw-offset conditions. *Wind Energy Science*, 3(1):173–189, 2018.

[4] B. L. Ennis, J. R. White, and J. A. Paquette. Wind turbine blade load characterization under

yaw offset at the swift facility. In *Journal of Physics: Conference Series*, volume 1037, page 052001. IOP Publishing, 2018.

[5] K. S. Heck, H. M. Johlas, and M. F. Howland. Modelling the induction, thrust and power of a yaw-misaligned actuator disk. *Journal of Fluid Mechanics*, 959:A9, 2023.

[6] C. Jané-Ippel, N. Bempedelis, R. Palacios, and S. Laizet. Bayesian optimisation of a two-turbine layout around a 2D hill using Large Eddy Simulations. In preparation, 2024.

[7] C. R. Shapiro, D. F. Gayme, and C. Meneveau. Modelling yawed wind turbine wakes: a lifting line approach. *Journal of Fluid Mechanics*, 841:R1, 2018.

[8] G. A. Speakman, M. Abkar, L. A. Martínez-Tossas, and M. Bastankhah. Wake steering of multirotor wind turbines. *Wind Energy*, 24(11):1294–1314, 2021.

[9] R. J. A. M. Stevens, D. F. Gayme, and C. Meneveau. Large eddy simulation studies of the effects of alignment and wind farm length. *Journal of Renewable and Sustainable Energy*, 6(2), 2014.